# Brain micro-inflammation at specific vessels dysregulates organ-homeostasis via the activation of a new neural circuit

Yasunobu Arima[1†], Takuto Ohki[1†], Naoki Nishikawa[1,2†], Kotaro Higuchi[1†], Mitsutoshi Ota[1], Yuki Tanaka[1], Junko Nio-Kobayashi[3], Mohamed Elfeky[1,4], Ryota Sakai[5], Yuki Mori[6], Tadafumi Kawamoto[7], Andrea Stofkova[1], Yukihiro Sakashita[1], Yuji Morimoto[2], Masaki Kuwatani[8], Toshihiko Iwanaga[3], Yoshichika Yoshioka[6], Naoya Sakamoto[8], Akihiko Yoshimura[5], Mitsuyoshi Takiguchi[9], Saburo Sakoda[10], Marco Prinz[11,12], Daisuke Kamimura[1], Masaaki Murakami[1]*

[1]Division of Psychoimmunology, Institute for Genetic Medicine and Graduate School of Medicine, Hokkaido University, Sapporo, Japan; [2]Department of Anesthesiology and Critical Care Medicine, Graduate School of Medicine, Hokkaido University, Sapporo, Japan; [3]Laboratory of Histology and Cytology, Graduate School of Medicine, Hokkaido University, Sapporo, Japan; [4]Department of Biochemistry, Faculty of Veterinary Medicine, Alexandria University, Behera, Egypt; [5]Department of Microbiology and Immunology, Keio University School of Medicine, Tokyo, Japan; [6]Laboratory of Biofunctional Imaging, WPI Immunology Frontier Research Center, Osaka University, Osaka, Japan; [7]Radioisotope Research Institute, Department of Dental Medicine, Tsurumi University, Yokohama, Japan; [8]Department of Gastroenterology and Hepatology, Graduate School of Medicine, Hokkaido University, Sapporo, Japan; [9]Laboratory of Veterinary Internal Medicine, Department of Veterinary Clinical Sciences, Graduate School of Veterinary Medicine, Hokkaido University, Sapporo, Japan; [10]Department of Neurology, National Hospital Organization Toneyama National Hospital, Osaka, Japan; [11]Institute of Neuropathology, Faculty of Medicine, University of Freiburg, Freiburg im Breisgau, Germany; [12]BIOSS Centre for Biological Signalling Studies, University of Freiburg, Freiburg im Breisgau, Germany

*For correspondence: murakami@igm.hokudai.ac.jp

†These authors contributed equally to this work

Competing interests: The authors declare that no competing interests exist.

**Abstract** Impact of stress on diseases including gastrointestinal failure is well-known, but molecular mechanism is not understood. Here we show underlying molecular mechanism using EAE mice. Under stress conditions, EAE caused severe gastrointestinal failure with high-mortality. Mechanistically, autoreactive-pathogenic CD4+ T cells accumulated at specific vessels of boundary area of third-ventricle, thalamus, and dentate-gyrus to establish brain micro-inflammation via stress-gateway reflex. Importantly, induction of brain micro-inflammation at specific vessels by cytokine injection was sufficient to establish fatal gastrointestinal failure. Resulting micro-inflammation activated new neural pathway including neurons in paraventricular-nucleus, dorsomedial-nucleus-of-hypothalamus, and also vagal neurons to cause fatal gastrointestinal failure. Suppression of the brain micro-inflammation or blockage of these neural pathways inhibited the gastrointestinal failure. These results demonstrate direct link between brain micro-inflammation and fatal gastrointestinal disease via establishment of a new neural pathway under stress. They further

suggest that brain micro-inflammation around specific vessels could be switch to activate new neural pathway(s) to regulate organ homeostasis.

## Introduction

Multiple sclerosis (MS) is an autoreactive helper T cell (CD4+ T cell)-mediated autoimmune disease in the central nervous system (CNS) (*Sawcer et al., 2011*). It can be divided into at least four sub-groups: (1) relapse-remitting MS (over 80% patients), (2) secondary progressive MS (the late phase of relapse-remitting type), (3) primary progressive MS (about 10% patients), and (4) relapse-progressing MS (about 5% patients), all of which are characterized by chronic inflammation in the CNS (*Steinman, 2009*). MS inflammatory lesions consist of various immune cells, including CD4+ T cells and MHC class II+CD11b+ monocytes/macrophages, and subsequent loss of neurological function in the CNS due to the disruption of neural circuits that regulate various organ functions (*Steinman, 2014*). Indeed, several complications including mental illnesses and gastrointestinal failures are associated with MS (*Goldman Consensus Group, 2005*; *Gupta et al., 2005*; *Kimura et al., 2000*; *Marrie et al., 2015*; *Pokorny et al., 2007*; *Rang et al., 1982*; *Sadovnick et al., 1989*). To understand the molecular mechanisms involved in the development of MS, experimental autoimmune encephalomyelitis (EAE) mice and rats have been designed (*Miller et al., 1995*; *Steinman, 2009*). EAE development is dependent on autoreactive helper T cells (pathogenic CD4+ T cells), particularly Th1 and Th17 cells and many immune cells are accumulated at the affected sites such as CD8+ T cells, macrophages, B cells, and neutrophils. (*Arima et al., 2012*, *2015*; *Jäger et al., 2009*). To develop EAE, CNS autoantigens including myelin oligodendrocyte glycoprotein (MOG), myelin basic protein (MBP), and proteolipid protein (PLP) are immunized (*Rangachari and Kuchroo, 2013*). We sorted CD4+ T cells from mice EAE induced with a MOG peptide plus complete Freund's adjuvant (CFA) (active EAE model) and stimulated the cells with a MOG peptide in the presence of MHC class II+ antigen presenting cells and cytokines in vitro followed by intravenously injecting them into wild type C57BL/6 mice (the transfer EAE model) (*Arima et al., 2012*, *2015*; *Rangachari and Kuchroo, 2013*). Therefore, the transfer EAE model is completely dependent on MOG-specific CD4+ T cells (pathogenic CD4+ T cells) including Th17 and Th1 populations, while it is not dependent on CD8+ T cells because of no CD8+ T cells in the transferred pathogenic T cell population. Importantly, there are many papers that describe the involvement of not only IL-17A but also IFN-γ particularly from pathogenic CD4+ T cells including activated Th1 and Th17 cells for the development of EAE models including the transfer EAE ones (*Duhen et al., 2013*; *El-behi et al., 2010*; *Goverman, 2009*; *Jäger et al., 2009*; *Ottum et al., 2015*). Using this transfer EAE model, we previously identified several specific sensory-sympathetic connections generate immune cell-gateways at specific vessels in the blood-brain barrier through which immune cells, including pathogenic CD4+ T cells, migrate to the CNS as the gateway reflex in the research field of the neural signal system (*Arima et al., 2012*, *2015*; *Chavan et al., 2017*; *Mori et al., 2014*; *Pavlov and Tracey, 2017*; *Sabharwal et al., 2014*; *Tracey, 2012*, *2016*).

Stress conditions can cause gastrointestinal diseases via the brain-gut axis. Animal models have shown that this axis involves interactions between neural components, including the autonomic nervous system, the central nervous system, the stress system such as the hypothalamic-pituitary-adrenal axis, and the corticotropin-releasing factor system, and intestinal factors such as the intestinal barrier, the luminal microbiota, and the intestinal immune response (*Allen et al., 2014*; *Bonaz and Bernstein, 2013*; *Caso et al., 2008*; *Stengel and Taché, 2009*). However, details about the underlying molecular mechanisms, including the relationship between each of these components, are lacking.

In the present paper, we report a relationship between brain micro-inflammation and fatal gastrointestinal failure. We demonstrate that brain micro-inflammation at blood vessels specific in the boundary area of the third ventricle region, thalamus, and dentate gyrus is established by pathogenic CD4+ T cell transfer under stress conditions and induces fatal gastrointestinal failure via a paraventricular nucleus (PVN)/dorsomedial nucleus of the hypothalamus (DMH)-vagal pathway. Mechanistic analysis suggested that the gateway reflex triggered by stress-mediated PVN activation induced the accumulation of immune cells, including pathogenic CD4+ T cells and MHC class

IIhiCD11b+ cells, at these specific vessels in a manner dependent on CCL5. The resulting brain micro-inflammation enhanced a new neural pathway including neurons in the PVN and DMH, and vagal nerve-mediated gastrointestinal failure via micro-inflammation-dependent ATP. Accordingly, blockade of the regional brain micro-inflammation and establishment of the new neural pathway inhibited the gastrointestinal pathology. These results suggest that brain micro-inflammation at specific vessels is a risk for severe gastrointestinal failure via the PVN/DMH-vagal neural pathway, indicating a direct molecular link between brain micro-inflammation and organ dysfunction.

## Results

### Stress conditions cause a severe phenotype of EAE

To examine the impact of stress conditions in a transfer EAE model, we first employed a sleep disorder model, in which continuous stress is imposed on mice on a free rotation wheel for 2 days by the perpetual avoidance of water (*Miyazaki et al., 2013*; *Oishi et al., 2014*). In the typical course of the transfer EAE model, a loss of tonicity in the tail tip and ascending paralysis follows immune cell accumulation at the lumbar (L)5 dorsal vessels after the intravenous transfer of pathogenic CD4+ T cells (*Arima et al., 2012*). Pathogenic CD4+ T cell transfer under the stress condition of the current study caused a severe phenotype including sudden death (*Figure 1A*). On the other hand, either stress or pathogenic CD4+ T cell transfer alone, or stress with the transfer of activated CD4+ T cells specific for irrelevant antigens such as ovalbumin or a retinal autoantigen, interphotoreceptor retinoid-binding protein (IRBP), did not cause the severe phenotype, although the retina-specific activated CD4+ T cells accumulated in the eyes (*Figure 1A and B* and *Figure 1—figure supplement 1A, B and C*). Blood aldosterone and cortisol levels were significantly higher in mice with stress even without pathogenic CD4+ T cell transfer (*Figure 1—figure supplement 2A and B*), suggesting stress-mediated activation of the hypothalamic-pituitary-adrenal axis. Another chronic stress model, in which mice were reared in a cage with wet bedding after pathogenic CD4+ T cell transfer, showed similar results (*Figure 1C and D*). However, transient stresses, such as immobilization stress and forced swimming (20 min/day, 10 days), did not (*Arima et al., 2015*). These results demonstrate that chronic stress conditions cause a severe and atypical phenotype of EAE, which might represent progressive MS.

### Gastrointestinal failure is induced by pathogenic CD4+ T cell transfer under stress

We next investigated how pathogenic CD4+ T cell transfer under stress causes the severe phenotype. We noticed that mice induced with pathogenic CD4+ T cell transfer under stress, but not either treatment only, excreted bloody stools (*Figure 2A*). Consistently, blood hematocrit levels decreased (*Figure 2B*). We found the blood content came from the stomach and upper level of the intestine (*Figure 2C*). Focal bleeding lesions in the stomach were detected by histopathological analysis (*Figure 2D*, black spots), and tissue distraction including epithelial necrosis was evident particularly in the stomach and upper level of the intestine (*Figure 2E*). It has been reported that under very strong stress, the activation of p38 and MAPKAPK2 contributes to stomach epithelial destruction under muscarinic M3 receptor signaling via acetyocholine derived from vagus nerve (*Debas and Carvajal, 1994*; *Jia et al., 2007*; *Uwada et al., 2017*). These reports further showed that p38 inhibitors could suppress the stomach epithelial destruction under very strong stress. We investigated the activation status of p38 and MAPKAPK2 in stomach isolated from mice with pathogenic CD4+ T cell transfer under our stress condition and detected phosphorylated p38 and phosphorylated MAPKAPK2 in the gastric mucosa (*Figure 2—figure supplement 1*). These results suggested tissue damage in the stomach and the upper level of the intestine due to p38 and MAPKAPK2 activation in mucosal tissues. Consistent with dysregulation of gastrointestinal region, the treatment with a proton pump inhibitor, lansoprazol, suppressed the severe phenotype in mice with pathogenic CD4+ T cell transfer under stress condition (*Figure 2F*). Furthermore, increase of plasma potassium levels, most likely following the gastrointestinal bleeding, was observed in mice with pathogenic CD4+ T cell transfer under stress condition (*Figure 2G*).

Heart failure was detected in mice with pathogenic CD4+ T cell transfer under stress condition by a cardiac electrocardiogram system (*Figure 2—figure supplement 2*), suggesting a heart failure at least partially mediated by high potassium ions. These results suggested that fatal gastrointestinal

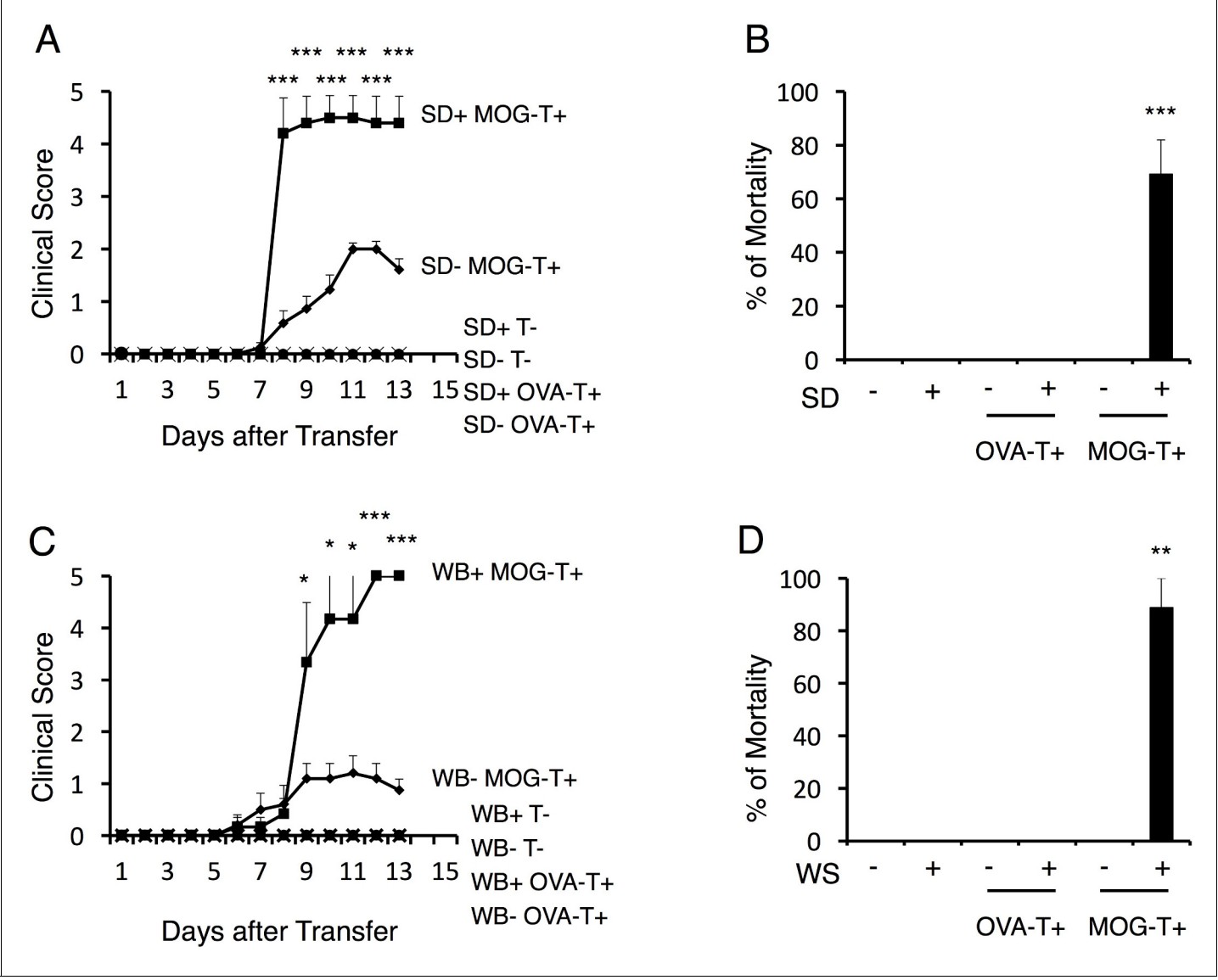

**Figure 1.** Stress conditions develop a severe phenotype of EAE. (**A**) Clinical scores of mice with no treatment (SD- T-), stress condition (sleep disorder (SD) stress) only (SD+ T-), OVA-specific CD4+ T cell transfer only (SD- OVA-T+), OVA-specific CD4+ T cell transfer under stress condition (SD+ OVA-T +), MOG-pathogenic CD4+ T cell transfer only (SD- MOG-T+), and MOG-pathogenic CD4+ T cell transfer under stress condition (SD+ MOG-T+) (n = 3–5 per group). (**B**) Percentages of mortality of mice with no treatment (SD- T-), stress condition only (SD+ T-), OVA-specific CD4+ T cell transfer only (SD- OVA-T+), OVA-specific CD4+ T cell transfer only (SD+ OVA-T+), MOG-pathogenic CD4+ T cell transfer only (SD- MOG-T+), and MOG-pathogenic CD4+ T cell transfer under stress condition (SD+ MOG-T+) 10 days after pathogenic CD4+ T cell transfer (n = 3–5 per group). (**C**) Clinical scores of mice with no treatment (WB- T-), stress condition (wet bedding (WB) stress) only (WB+ T-), OVA-specific CD4+ T cell transfer only (WB- OVA-T +), OVA-specific CD4+ T cell transfer only (WB+ OVA-T+), MOG-pathogenic CD4+ T cell transfer only (WB- MOG-T+), and MOG-pathogenic CD4+ T cell transfer under stress condition (WB+ MOG-T+) (n = 5 per group). (**D**) Percentages of mortality of mice with no treatment (WB- T-), stress condition (wet bedding stress) only (WB+ T-), OVA-specific CD4+ T cell transfer only (WB- OVA-T+), OVA-specific CD4+ T cell transfer only (WB+ OVA-T+), MOG-pathogenic CD4+ T cell transfer only (WB- MOG-T+), and MOG-pathogenic CD4+ T cell transfer under stress condition (WB+ MOG-T+) 10 days after transfer (n = 3–5 per group). Mean scores ± SEM are shown. Statistical significance was determined by ANOVA tests. Statistical significance is denoted by asterisks (*p<0.05, **p<0.01, ***p<0.001). Experiments were performed at least three times; representative data are shown.

The following figure supplements are available for figure 1:

**Figure supplement 1.** Autoreactive CD4+ T cells against a retinal antigen IRBP did not cause the severe phenotype under stress condition.

**Figure supplement 2.** Blood aldosterone and cortisol levels were significantly higher in mice under stress condition independent of pathogenic CD4+ T cell transfer.

*Figure 1 continued on next page*

Figure 1 continued

Figure supplement 3. The mortality was not affected by corticosteroid receptor antagonist treatment in cytokine-microinjected mice under stress condition.

failure is induced by pathogenic CD4+ T cell transfer under stress condition, which unexpectedly revealed an unpredicted association between CNS micro-inflammation and organ homeostasis including gastrointeste and heart.

## Brain micro-inflammation at specific vessels is developed in a manner dependent on chemokines and inflammatory cytokines after pathogenic CD4+ T cell transfer under stress condition

Because the clinical symptoms were quite different between mice having pathogenic CD4+ T cell transfer with or without stress, we hypothesized that the positions of the vessels in the CNS where pathogenic CD4+ T cells initially accumulated were different. Donor pathogenic CD4+ T cells and MHC class II+ cells first accumulate at the L5 cord within 5 days after pathogenic CD4+ T cell transfer via the gravity-gateway reflex as described previously (*Arima et al., 2012*; *Sabharwal et al., 2014*) (*Figure 3A*). In the present work, chronic stress caused pathogenic CD4+ T cells and MHC class II+ cells to accumulate at specific vessels of the boundary region of the third ventricle region, thalamus, and dentate gyrus, but not in the L5 cord (*Figure 3B*). These data suggested that the L5 cord vessels are no longer used as a gateway for immune cells toward the CNS under chronic stress conditions. Furthermore, EAE mice under stress had no paralyzed tail, uneven gait, or paralyzed rear leg, which are all normally observed in EAE mice without stress. These data are consistent with the L5 cord vessels not acting as a gateway for immune cells under chronic stress conditions. We also found that pathogenic CD4+ T cells and MHC class II+ cells accumulated comparably at the brain specific vessels redardless of lansoprazol treatment, which suppressed gastrointestine dysregulation (see Fiugre 2F), in mice with pathogenic CD4+ T cell transfer under stress condition (*Figure 3—figure supplement 1A, B and C*). These data suggested that chronic stress changes the position of the gateway for immune cells from L5 cord vessels to brain specific vessels, while pathogenic CD4+ T cells are activated and acculumated at the specific vessels even after lansoprazol treatment. In addition, we showed that lansoprazol treatment did not affect the accumulation of immune cells in L5 and EAE symptoms in mice without stress (*Figure 3—figure supplement 1D*), confirming that lansoprazol has no or minimun affect on pathogenic T cells. All these data suggested that lansoprazol treatment does not affect the EAE induction and that the stress-gateway reflex alters the gateway of pathogenic CD4+ T cells. Flow cytometry analysis confirmed an abundance of immune cell accumulation including pathogenic CD4+ T cells and MHC class IIhiCD11b+ cells in the hippocampus and interbrain area, where the specific vessels were localized (*Figure 3C*). Td-tomato labeling of microglia cells, which we used previously (*Arima et al., 2015*), revealed that MHC class IIhiCD11b+ cells originated from the peripheral organs rather than resident microglia cells (*Figure 3D*), suggesting that the MHC class IIhiCD11b+ cells that accumulated at the specific vessels were activated monocytes from the peripheral organs. Moreover, we found that various immune cells, including CD8+ T cells, B cells, NK cells, and neutrophils, had also accumulated at the specific vessels (*Figure 3—figure supplement 2*). Thus, the stress condition induced brain micro-inflammation at specific vessels of the boundary area of the third ventricle region, thalamus, and dentate gyrus in the transfer EAE model.

We next sought to identify the chemokine(s) responsible for the immune cell accumulation at the brain micro-inflammation and tested several neutralizing antibodies against chemokines particularly known to recruit CD4+ T cells and MHC class II+ antigen presenting cells, because the activation of pathogenic CD4+ T cells in the affected sites are critical for inflammation in transfer EAE. Blockade of CCL5, but not chemokines including CCL2, CX3CL1, significantly suppressed the accumulation of pathogenic CD4+ T cells and MHC class IIhiCD11b+ cells at the specific vessels (*Figure 4A and B* and *Figure 4—figure supplement 1*) and reduced clinical scores and mortality (*Figure 4C*). Interestingly, the stress condition even without pathogenic CD4+ T cell transfer induced CCL5 expression at

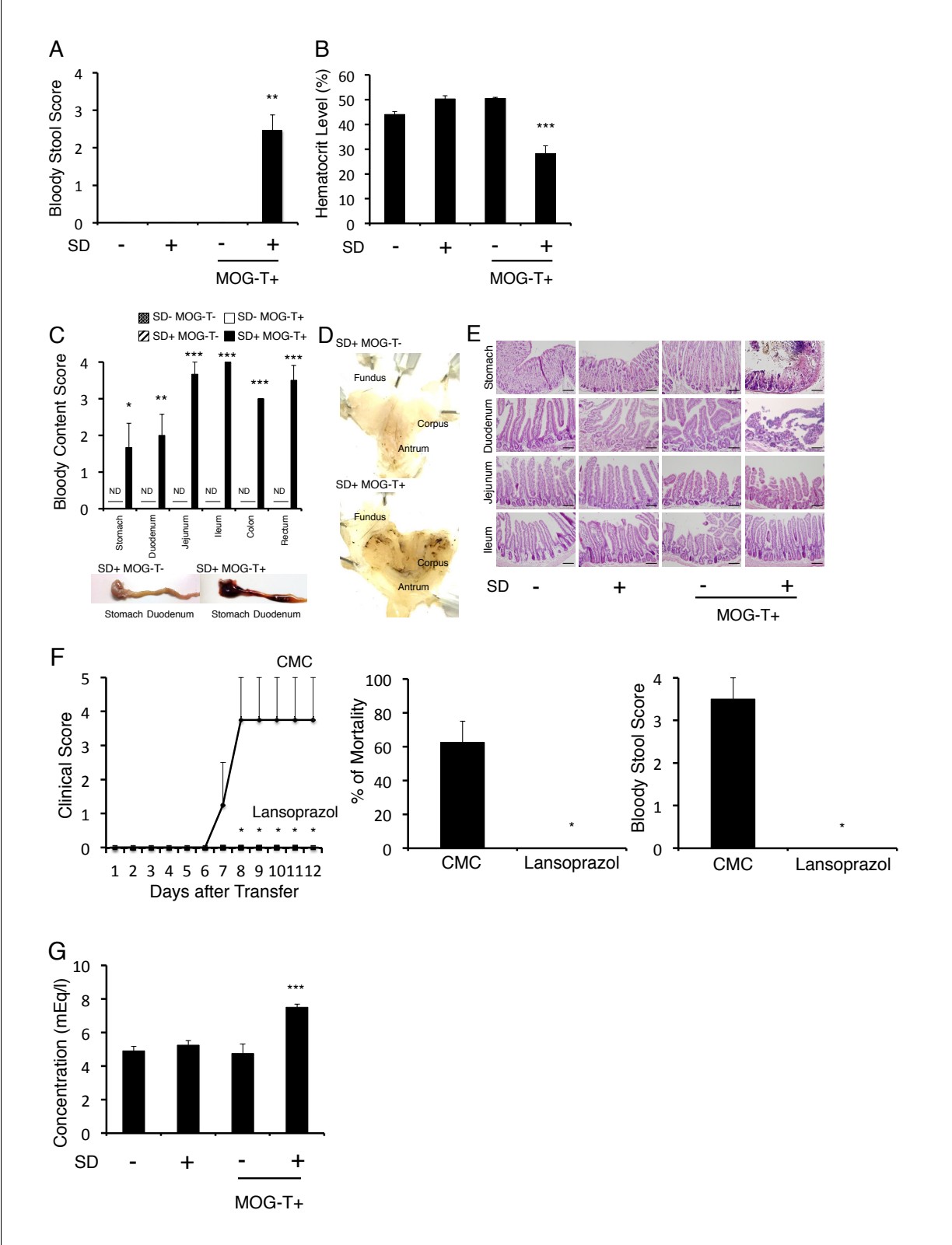

**Figure 2.** Gastrointestinal failure is induced after pathogenic CD4+ T cell transfer under stress condition. (**A**) Bloody stool scores by a fecal occult blood test of mice with no treatment (SD- MOG-T-), stress condition only (SD+ MOG-T-), pathogenic CD4+ T cell transfer only (SD- MOG-T+), and pathogenic CD4+ T cell transfer under stress condition (SD+ MOG-T+) 10 days after pathogenic CD4+ T cell transfer (n = 3–4 per group). (**B**) Blood hematocrit levels in mice with no treatment (SD- MOG-T-), stress condition only (SD+ MOG-T-), pathogenic CD4+ T cell transfer only (SD- MOG-T+),
*Figure 2 continued on next page*

*Figure 2 continued*

and pathogenic CD4+ T cell transfer under stress condition (SD+ MOG-T+) 10 days after pathogenic CD4+ T cell transfer (n = 3–4 per group). (C) Bloody content scores by a fecal occult blood test of mice with no treatment (SD- MOG-T-), stress condition only (SD+ MOG-T-), pathogenic CD4+ T cell transfer only (SD- MOG-T+), and pathogenic CD4+ T cell transfer under stress condition (SD+ MOG-T+) 10 days after pathogenic CD4+ T cell transfer. Stomach and small and large intestines were collected after perfusion (n = 3–4 per group). ND, not detected. (D) Pathological analysis of stomach in mice with stress condition only (SD+ MOG-T-) and pathogenic CD4+ T cell transfer under stress condition (SD+ MOG-T+) 10 days after pathogenic CD4+ T cell transfer (n = 3 per group). (E) Pathological analysis of stomach and small intestine of mice with no treatment (SD- MOG-T-), stress condition only (SD+ MOG-T-), pathogenic CD4+ T cell transfer only (SD- MOG-T+), and pathogenic CD4+ T cell transfer under stress condition (SD+ MOG-T+) 10 days after pathogenic CD4+ T cell transfer (n = 3 per group). Scale bars represent 100 μm. (F) Clinical scores, percentages of mortality, and bloody stool scores of mice with or without lansoprazol treatment after pathogenic CD4+ T cell transfer and stress condition (n = 3–5 per group). Percentages of mortality and bloody stool score were evaluated 10 days after pathogenic CD4+ T cell transfer. (G) Plasma potassium levels in mice with no treatment (SD- MOG-T-), stress condition only (SD+ MOG-T-), pathogenic CD4+ T cell transfer only (SD- MOG-T+), and pathogenic CD4+ T cell transfer under stress condition (SD+ MOG-T+) 10 days after pathogenic CD4+ T cell transfer (n = 4–5 per group). Mean scores ± SEM are shown. Statistical significance was determined by ANOVA tests. Statistical significance is denoted by asterisks (*p<0.05, **p<0.01, ***p<0.001). ND, not detected. Experiments were performed at least three times; representative data are shown.

The following figure supplements are available for figure 2:

**Figure supplement 1.** The phosphorylated p38 and phosphorylated MAPKAPK 2 levels were significantly higher in gastric mucosa of mice with pathogenic CD4+ T cell transfer under stress condition.

**Figure supplement 2.** Heart failure was induced in mice with pathogenic CD4+ T cell transfer under stress condition.

---

the specific blood vessels, but acute stress models did not. (*Figure 4D* and *Figure 4—figure supplement 2*). These data suggest that chronic stress but not acute stress is capable to induce the nerve activation in PVN that elicits CCL5 expression at the specific vessels.

Moreover, neutralization of the Th1 cytokine, interferon-gamma (IFN-γ), or that of the Th17 cytokine, IL-17A, also inhibited the brain micro-inflammation at the specific vessels (*Figure 4E*). Importantly, the accumulation of IL-17A- or IFN-γ-deficient pathogenic CD4+ T cells was significantly reduced at the specific blood vessels under stress conditions and resulted in less mortality (*Figure 4—figure supplement 3*). These data suggested that both IFN-γ and IL-17A from pathogenic CD4+ T cells are necessary for the accumulation at specific vessels and the severe phenotypes. Consistent with these results, prominent inhibitory effects on the disease development were observed with combined neutralization of IL-17A and IFN-γ(*Figure 4F*). These results suggested that stress-mediated CCL5 expression at the specific vessels induces the accumulation of pathogenic CD4+ T cells, particularly Th17 and Th1 cells and MHC class IIhiCD11b+ cells to establish brain micro-inflammation, which is critical for the development of fatal gastrointestinal failure. We call this phenomenon the stress-gateway reflex.

## Brain micro-inflammation at specific vessels is sufficient to induce fatal gastrointestinal failure under stress condition

We next investigated whether brain micro-inflammation at the specific vessels is sufficient to develop intestinal failure in mice under stress. To answer this question, we directly microinjected pathogenic CD4+ T cells plus MOG-pulsed DC or inflammatory cytokines, such as IFN-γ plus IL-17A or IL-6 plus IL-17A, either of which is known to be expressed by pathogenic CD4+ T cells and enhance chemokine expression at specific vessels in gateway reflexes previously identified (*Arima et al., 2012*; *Sabharwal et al., 2014*), at specific vessels of the boundary region of the third ventricle region, thalamus, and dentate gyrus under stress condition. We found that brain micro-inflammation induced by these treatments developed severe gastrointestinal failure and affected mortality (*Figure 4G*). Furthermore, we investigated the mortality of mice with microinjections of cytokines at the specific vessels under stress condition in the presence or absence of anti-CCL5 antibody treatment and found anti-CCL5 antibody treatment had no significant effect (*Figure 4—figure supplement 4*). This result suggested that CCL5 mainly contributes to the accumulation of immune cells including pathogenic CD4+ T cells at the specific vessels, while the effect of cytokine injection is CCL5-independent. These results suggest that brain micro-inflammation at the specific vessels, which is triggered by

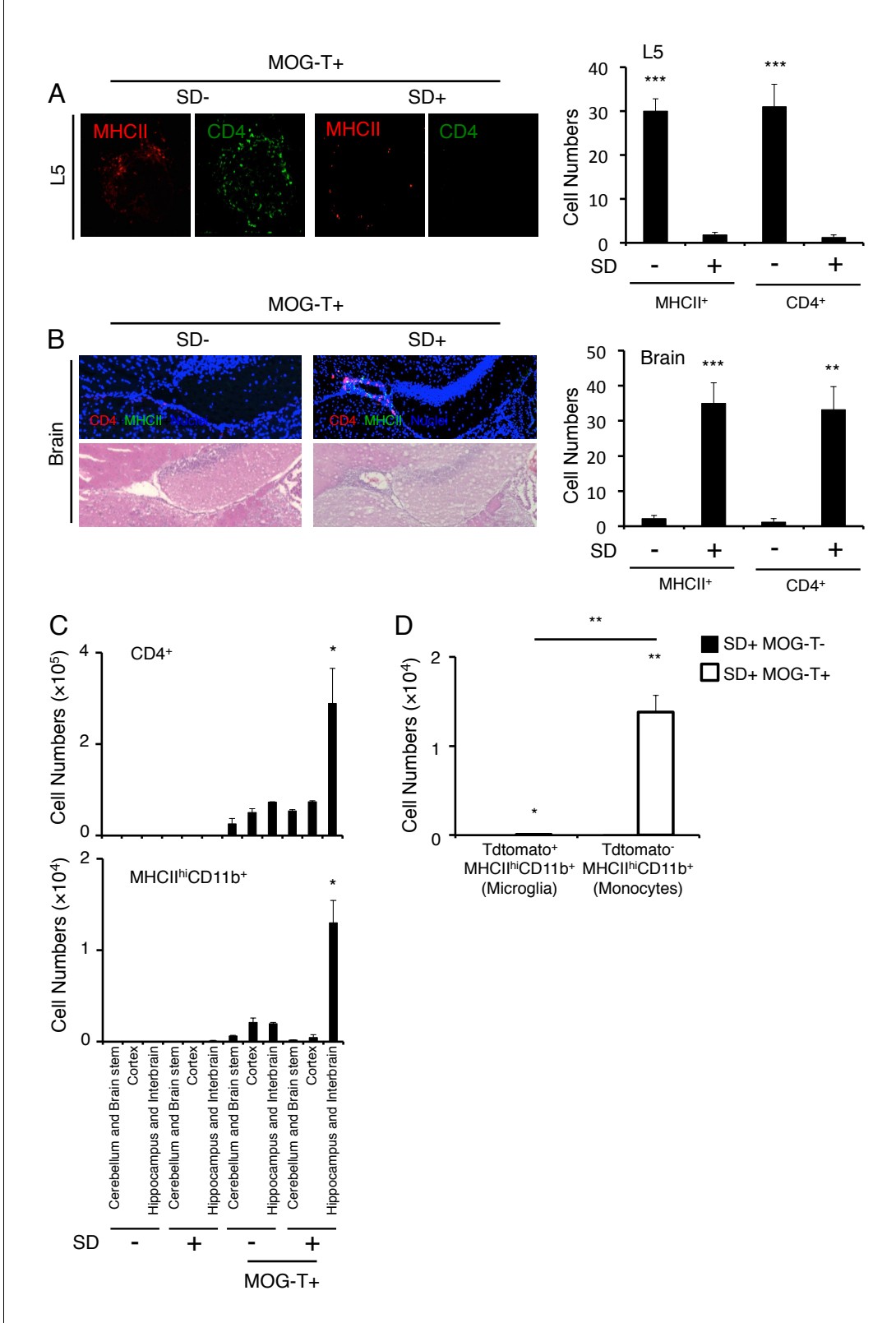

**Figure 3.** Brain micro-inflammation was developed at specific vessels of the boundary area of the third ventricle region, thalamus, and dentate gyrus after pathogenic CD4+ T cell transfer under stress condition. (**A**) Immunohistochemical staining for CD4 and MHC class II in the L5 cord of mice in the presence or absence of stress condition (SD) 5 days after pathogenic CD4+ T cell transfer (n = 3 per group). (right) Quantification of the histological analysis. Number of cells per picture (10x). (**B**) Immunohistochemical staining for CD4 and MHC class II at specific vessels of the boundary area of the

*Figure 3 continued on next page*

*Figure 3 continued*

third ventricle region, thalamus, and dentate gyrus in mice with or without stress condition (SD) in the presence of T cell transfer (n = 3 per group). (right) Quantification of the histological analysis. Number of cells per picture (10x). (C) Numbers of CD4+ T cells and MHC class IIhiCD11b+ cells in different brain regions of mice with no treatment (SD- MOG-T-), stress condition only (SD+ MOG-T-), pathogenic CD4+ T cell transfer only (SD- MOG-T +), and pathogenic CD4+ T cell transfer under stress condition (SD+ MOG-T+) 10 days after pathogenic CD4+ T cell transfer (n = 3–4 per group). Mean scores ± SEM are shown. (D) Numbers of microglia and monocytes in the hippocampi and interbrains area of tamoxifen-treated CX3CR1 CreER ROSA26-TdTomato mice 10 days after pathogenic CD4+ T cell transfer (n = 4–5 per group). Mean scores ± SEM are shown. Statistical significance was determined by ANOVA tests. Statistical significance is denoted by asterisks (*p<0.05, **p<0.01, ***p<0.001). Experiments were performed at least three times; representative data are shown.

The following figure supplements are available for figure 3:

**Figure supplement 1.** Pathogenic CD4+ T cells and MHC class II+ cells accumulated at the specific vessels but not L5 cord with or without lansoprazol treatment.

**Figure supplement 2.** Various immune cells had accumulated at the specific vessels in mice after pathogenic CD4+ T cell transfer under stress condition.

**Figure supplement 3.** The chronic stress condition reduced CCL20 expression at L5 dorsal blood vessels and cfos expression in L5 DRG.

CCL5-mediated pathogenic CD4+ T cell accumulation, induces regional cytokine increment followed by severe gastrointestinal failure under stress condition.

## The PVN-meditated sympathetic pathway is involved in the development of brain micro-inflammation at specific vessels under stress condition

We next investigated how pathogenic CD4+ T cell transfer under stress condition establishes the initial gateway for immune cells at specific vessels of the boundary area of the third ventricle region, thalamus, and dentate gyrus. We previously found that the gateway reflexes are dependent on sympathetic/noradrenergic pathways distributed at the target vessels (*Arima et al., 2012*, *2015*) and presently found that stress conditions increased sympathetic tone and serum aldosterone and cortisol levels (*Figure 1—figure supplement 2A and B*). Therefore, we performed chemical sympathectomy at the specific vessels. Microinjection of 6-hydroxydopamine (6-OHDA) at the specific vessels successfully depleted tyrosine hydroxylase (TH)+ sympathetic neurons and phospho-CREB signals that were induced by noradrenergic receptor signaling after sympathetic activation (*Figure 5—figure supplement 1*). 6-OHDA-mediated chemical sympathectomy suppressed immune cell accumulation at the specific vessels and the development of fatal gastrointestinal failure in mice that received pathogenic CD4+ T cell transfer under stress condition (*Figure 5A–5B*).

We then investigated how sympathetic activation at the specific vessels under stress condition is regulated. It is reported that PVN neurons include TH+ neurons are activated by several stresses, especially chronic ones (*Herman and Cullinan, 1997*; *Ramot et al., 2017*; *Shi et al., 2013*; *Ulrich-Lai and Herman, 2009*). Consistent with these results, we found that PVN TH+ neurons were specifically activated by stress condition even without EAE induction (*Figure 5C*). Moreover, the activation of PVN neurons, particularly TH$^{neg}$ neurons, was enhanced in mice with pathogenic CD4+ T cell transfer under stress condition (*Figure 5C*), suggesting the crosstalk between microinflammation at the specifc vessels and PVN. We further examined whether there is a direct neural connection between the specific vessels and the PVN. We found that injection of a retrograde neural tracer, cholera toxin B (CTB), at specific vessels of the boundary area of third ventricle region, thalamus, and dentate gyrus reached the PVN (*Figure 5D*). In addition, TH+ neurons at the PVN co-expressed noradrenaline transporter, but not dopamine transporter (*Figure 5—figure supplement 2*), suggesting that they secrete noradrenaline at the specific vessels, which we have shown to be important for chemokine induction (*Arima et al., 2012*, *2015*; *Sabharwal et al., 2014*). These results demonstrated that the PVN-sympathetic pathway is involved in the regional brain micro-inflammation and severe intestinal failure.

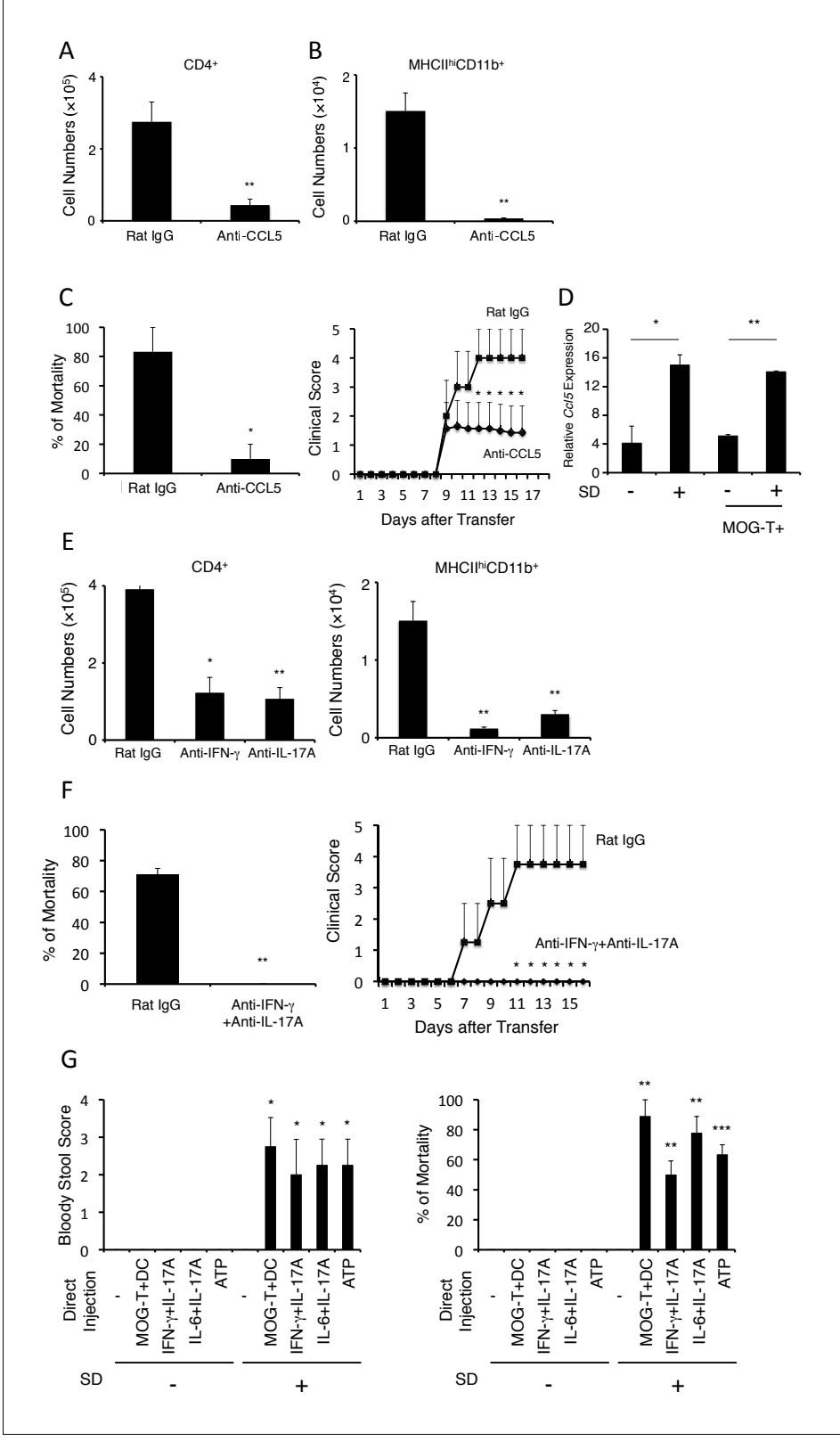

**Figure 4.** The development of brain micro-inflammation at specific vessels in mice with pathogenic CD4+ T cell transfer under stress condition is dependent on CCL5, IL-17, and IFN-γ. (**A and B**) Numbers of CD4+ T cells and MHC class IIhiCD11b+ cells in the hippocampi and interbrains of mice under stress condition in the presence or absence of anti-CCL5 antibody treatment 10 days after pathogenic CD4+ T cell transfer (n = 4–5 per group). (**C**)

*Figure 4 continued on next page*

*Figure 4 continued*

Percentages of mortality and clinical scores of mice under stress condition in the presence or absence of anti-CCL5 treatment (n = 3–5 per group). (**D**) CCL5 mRNA expression at specific vessels of the boundary area of the third ventricle region, thalamus, and dentate gyrus in mice with no treatment (SD- MOG-T-), stress condition only (SD+ MOG-T-), pathogenic CD4+ T cell transfer only (SD- MOG-T+), and pathogenic CD4+ T cell transfer under stress condition (SD+ MOG-T+) 4 days after pathogenic CD4+ T cell transfer (n = 3–5 per group). (**E**) Numbers of CD4+ T cells and MHC class IIhiCD11b+ cells in the hippocampi and interbrain area of mice under stress condition in the presence or absence of anti-IFN-γ antibody, anti-IL-17A antibody, or control 10 days after pathogenic CD4+ T cell transfer (n = 4–5 per group). (**F**) Percentages of mortality and clinical scores of mice with pathogenic CD4+ T cell transfer under stress condition in the presence or absence of anti-IFN-γ plus anti-IL-17A antibody treatment (n = 3–5 per group). (**G**) Bloody stool scores and percentages of mortality in mice under stress condition 2 days after microinjection of MOG-T plus DC, IFN-γ and IL-17A, or IL-6 and IL-17A at specific vessels of the boundary area of the third ventricle region, thalamus, and dentate gyrus (n = 3–5 per group). Mean scores ± SEM are shown. Statistical significance was determined by Student's t tests (A, B) and ANOVA tests (C-G). Statistical significance is denoted by asterisks (*p<0.05, **p<0.01). Experiments were performed at least three times; representative data are shown.

The following figure supplements are available for figure 4:

**Figure supplement 1.** Anti-CCL2 and Anti-CX3CL1 antibody had not suppressed the accumulation of pathogenic CD4+ T cells and MHC class IIhiCD11b+ cells at the specific vessels.

**Figure supplement 2.** The chronic stress condition induced CCL5 expression at the specific blood vessels but not acute stress condition.

**Figure supplement 3.** Pathogenic CD4+ T cells derived from IL-17A deficient or IFN-γ deficient mice inhibited the severe phenotypes.

**Figure supplement 4.** The mortality was not affected by anti-CCL5 antibody treatment in cytokines-microinjected mice under stress condition.

**Figure supplement 5.** CD11b+ cells isolated from mice with pathogenic CD4+ T cell transfer under stress condition have the potential of antigen presentation to CD4+ T cells without peptide addition.

## Brain micro-inflammation at the vessels under stress condition enhances activation of DMH neurons and develops severe gastrointestinal failure

We next searched for neural projections from the vessels of the boundary area of the third ventricle region, thalamus, and dentate gyrus. We found that an anterograde tracer, PHA-L, injected at these vessels mainly reached the DMH and to a lesser degree the PVN (*Figure 5E*). Consistent with this result, CTB injection at the DMH region reached the specific vessels (*Figure 5F*). Importantly, DMH neurons, particularly THneg ones, were highly activated after pathogenic CD4+ T cell transfer under stress condition (*Figure 5C*). These results suggested that regional brain micro-inflammation activates neurons mainly distributed in the DMH. Based on this finding, we propose that the brain micro-inflammation acts as a switch to establish new neural pathways including DMH that regulate gastrointestinal homeostasis under stress conditions.

Consistent with this notion, the injection of pathogenic CD4+ T cells plus MOG-pulsed DC or IL-17A plus IFN-γ cytokines at the specific vessels under stress condition enhanced neural activation of the PVN and DMH (*Figure 5G*). Furthermore, the ablation of PVN and DMH sites significantly inhibited brain micro-inflammation, gastrointestinal disease and mortality (*Figure 5H*). It is reported that the activation of DMH neurons is suppressed by GABA-A receptor activation (*Rusyniak et al., 2008*). Muscimol, GABA-A receptor agonist, injection at DMH suppressed gastrointestinal disease and mortality (*Figure 5I*). All these results suggested that DMH activation following the microinflammation at the specific vessels is critical for the development of fatal gastrointestinal disease.

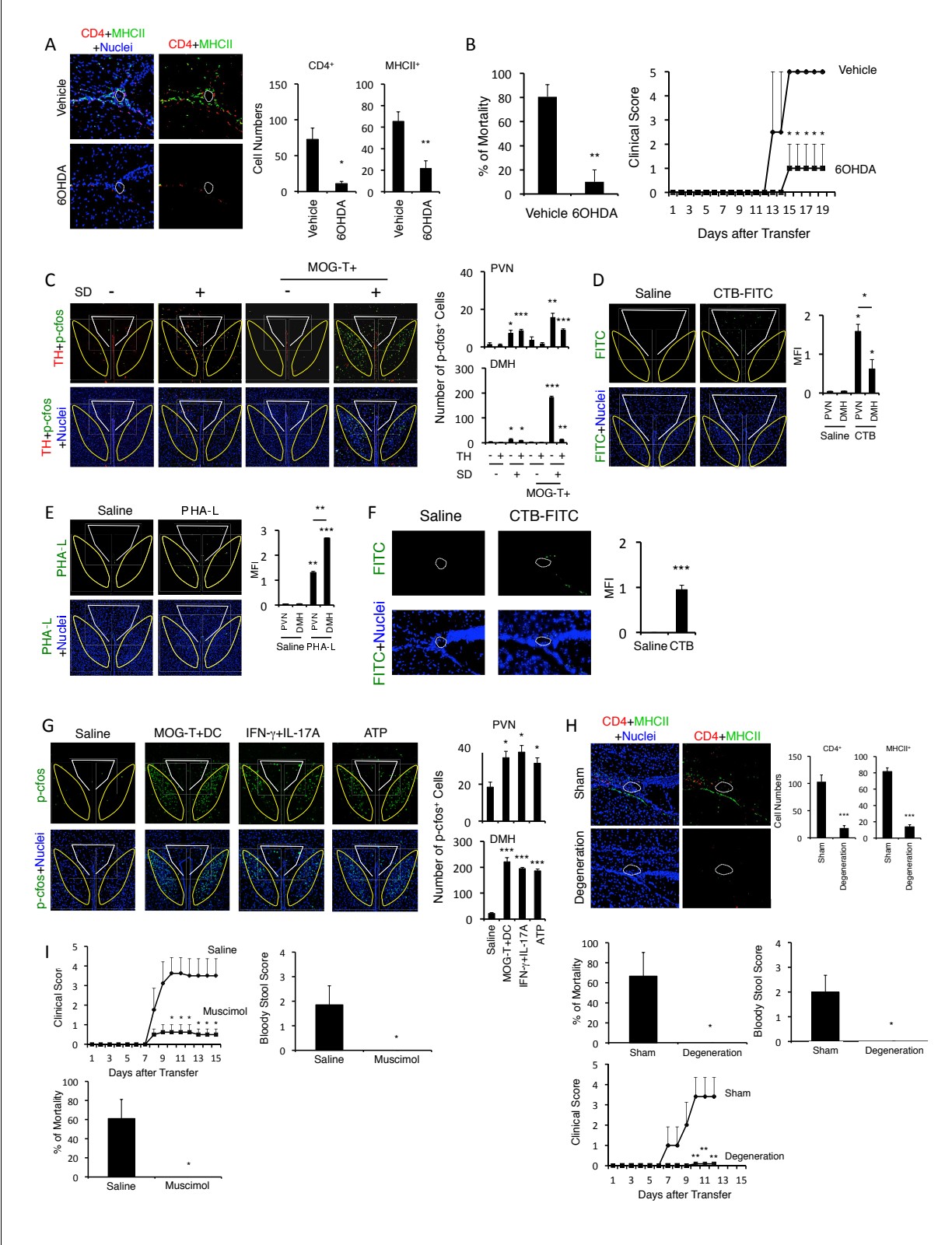

**Figure 5.** Neural activation of DMH following brain micro-inflammation is critical for the development of severe gastrointestinal failure. (**A**) Immunohistochemical staining for CD4 and MHC class II at specific vessels of the boundary area of the third ventricle region, thalamus, and dentate gyrus in mice under stress condition in the presence or absence of 6-OHDA microinjection at the specific vessels 10 days after pathogenic CD4+ T cell transfer. White polygon: the specific blood vessels of the boundary area of the third ventricle region, thalamus, and dentate gyrus. (right) Quantification

*Figure 5 continued on next page*

*Figure 5 continued*

of the histological analysis (n = 3 per group). Number of cells per picture (10x). (**B**) Percentages of mortality and clinical scores of mice in the presence or absence of microinjection of 6-OHDA at the specific vessels 10 days after pathogenic CD4+ T cell transfer (n = 3–5 per group). (**C**) Immunohistochemical staining for phospho-cfos and TH in the PVN and DMH of mice with no treatment (SD- MOG-T-), stress condition only (SD+ MOG-T-), pathogenic CD4+ T cell transfer only (SD- MOG-T+), and pathogenic CD4+ T cell transfer under stress condition (SD+ MOG-T+) 10 days after pathogenic CD4+ T cell transfer. White polygon: PVN, Yellow polygon: DMH. (right) Quantification of the histological analysis (n = 3–5 per group). Number of cells per picture (10x). (**D**) FITC-conjugated cholera toxin B in the PVN and DMH of C57BL/6 mice with or without microinjection of FITC-conjugated cholera toxin B at the specific vessels 5 days after pathogenic CD4+ T cell transfer. White polygon: PVN, Yellow polygon: DMH. (right) Quantification of the histological analysis (n = 3–5 per group). (**E**) PHA-L in the PVN and DMH of C57BL/6 mice with or without microinjection of PHA-L at the specific vessels 5 days after pathogenic CD4+ T cell transfer. White polygon: PVN, Yellow polygon: DMH. (right) Quantification of the histological analysis (n = 3–5 per group). (**F**) FITC-conjugated cholera toxin B at the specific vessels of C57BL/6 mice with or without microinjection of FITC-conjugated cholera toxin B in the DMH 5 days after pathogenic CD4+ T cell transfer (n = 3–5 per group). (**G**) Immunohistochemical staining for phosphor-cfos in the PVN and DMH of mice under stress condition 2 days after microinjection of MOG-T plus DC, IFN-γ plus IL-17A, or ATP at the specific vessels of the boundary area of the third ventricle region, thalamus, and dentate gyrus. (right) Quantification of the histological analysis (n = 3 per group). Number of cells per picture (10x). White polygon: PVN, Yellow polygon: DMH. (**H**) Immunohistochemical staining for CD4 and MHC class II at the specific vessels of the boundary area of the third ventricle region, thalamus, and dentate gyrus of mice in the presence or absence of PVN and DMH unilateral ablation 10 days after pathogenic CD4+ T cell transfer. (right) Quantification of the histological analysis (n = 3–4 per group). Number of cells per picture (10x). Percentages of mortality of mice under stress condition in the presence or absence of the unilateral ablation 10 days after pathogenic CD4+ T cell transfer (degeneration). Clinical scores are also shown (n = 4–5 per group). (**I**) Percentages of mortality and clinical scores of mice in the presence or absence of microinjection of muscimol at DMH 5 days after pathogenic CD4+ T cell transfer (n = 3–5 per group). Mean scores ± SEM are shown. Statistical significance was determined by Student's t tests (A, F) and ANOVA tests (B-E, G-I). Statistical significance is denoted by asterisks (*p<0.05; **p<0.01; ***p<0.001). Experiments were performed at least three times; representative data are shown.

The following figure supplements are available for figure 5:

**Figure supplement 1.** TH and pCREB signals were reduced in 6OHDA-mediated sympathectomized mice under stress.

**Figure supplement 2.** TH-positive neurons co-expressed noradrenaline transporter, but not dopamine transporter.

## ATP induced at the sites of brain micro-inflammation activates the DMH and severe gastrointestinal failure

ATP is a neurotransmitter and is expressed in several cells after cytokine stimulation (*Burnstock, 2006*). We found that the expression of ATP was enhanced after IL-17A plus IL-6 stimulation in endothelial cells in vitro (*Figure 6A*). Neurons in the DMH were highly activated by the microinjection of ATP at the specific vessels under stress condition (*Figure 5G*). Moreover, blockade of the ATP receptor, P2RX7, at the specific vessels by A438079, a selective antagonist, suppressed the neural activation and fatal gastrointestinal failure (*Figure 6B and C*). Thus, these results suggested that ATP is downstream of cytokine stimulation and acts as a neural stimulator. Moreover, we performed immunohistochemical staining results. Neurons (Neurofilament L-positive cells) around the specific vessels expressed P2RX7 and after cytokine or ATP stimulation, neurons in the DMH region also showed NFATc1 activation (*Figure 6—figure supplement 1*), which is a downstream signaling component of P2RX7 (*Grol et al., 2013*). Thus, we suggested that ATP can be sensed by neurons that connect specific vessels to the DMH region in a manner dependent on P2RX7 expression. In combination with other data in this paper, we propose the following pathogenic pathway: CCL5 (from endothelial cells at the specific vessels)→ IFN-γ/IL-17A/IL-6 (from pathogenic CD4+ T cells) → ATP (from endothelial cells at the specific vessels)→ neural activation in the DMH region. These results suggest that ATP at the sites of brain micro-inflammation activates the DMH to cause severe gastrointestinal failure.

## Enhanced activation of the vagal nerve is critical for the development of fatal gastrointestinal failure under stress condition

We next investigated whether vagal nerve activation is involved in the development of the severe gastrointestinal failure triggered by brain micro-inflammation, because (i) the status of PVN neurons affects the homeostasis of gastrointestinal organs via branches of the vagal nerves (*Ferguson et al., 1988*) and (ii) the DMH is connected to the vagal neurons at the dorsal motor vagal nucleus (DMX), where the nucleus of the vagal nerves are present (*Thompson et al., 1996*). We hypothesized that

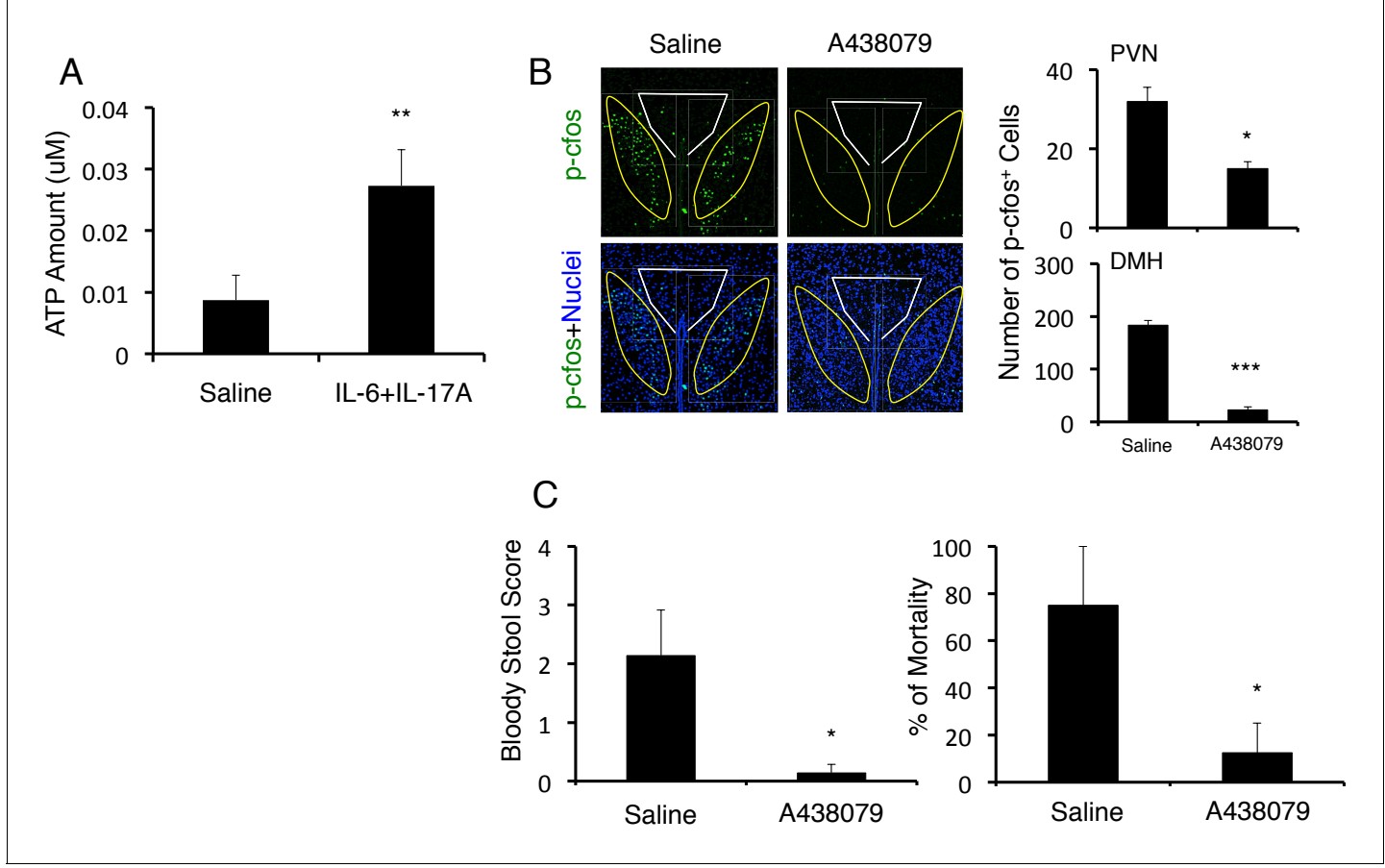

**Figure 6.** ATP induced at brain micro-inflammation sites causes severe gastrointestinal failure by activating DMH neurons. (**A**) ATP expression from BC1 cells after stimulation with IL-6 and IL-17A. (**B**) Immunohistochemical staining for phospho-cfos at the PVN and DMH of mice with microinjection of IL-6 plus IL-17A under stress condition in the presence or absence of A438079 at specific vessels of the boundary area of the third ventricle region, thalamus, and dentate gyrus 2 days after microinjection. (right) Quantification of the histological analysis (n = 3 per group). Number of cells per picture (10x). White polygon: PVN, Yellow polygon: DMH. (**C**) Bloody stool scores and percentages of mortality of mice with direct injection of IL-6 plus IL-17A under stress condition in the presence or absence of A438079 at the specific vessels 2 days after microinjection (n = 3–5 per group). Mean scores ± SEM are shown. Statistical significance was determined by Student's t tests (A, B) and ANOVA tests (C). Statistical significance is denoted by asterisks (*p<0.05; **p<0.01; ***p<0.001). Experiments were performed at least three times; representative data are shown.
The following figure supplement is available for figure 6:

**Figure supplement 1.** ATP activated the neurons in the DMH via P2RX7 receptor.

activation of the DMH and PVN enhances the activation of vagal neurons distributed in the stomach and upper level of the intestines. PHA-L after injection at the DMH reached the DMX as described (*Figure 7—figure supplement 1*). Moreover, we found that the DMX is highly activated after the injection of pathogenic CD4+ T cells with MOG-pulsed DC at the vessels of the boundary area of the third ventricle region, thalamus, and dentate gyrus under stress condition (*Figure 7A*). In addition, we found that Nucleus of the Tractus Solitaries (NTS), which is the main nucleus of afferent vagus nerves, was activated (*Figure 7A*), suggesting that afferent neuronal activation is induced by severe gastrointestinal failure, most likely in a manner dependent on micro-inflammation-related ATP in gastrointestinal regions. Finally, we found that vagotomy suppressed gastrointestinal failure and mortality in EAE mice under stress (*Figure 7B*). These results suggest that severe gastrointestinal failure is induced by enhanced neural activation via the DMH-vagal axis, which is activated by brain micro-inflammation at the specific vessels under stress.

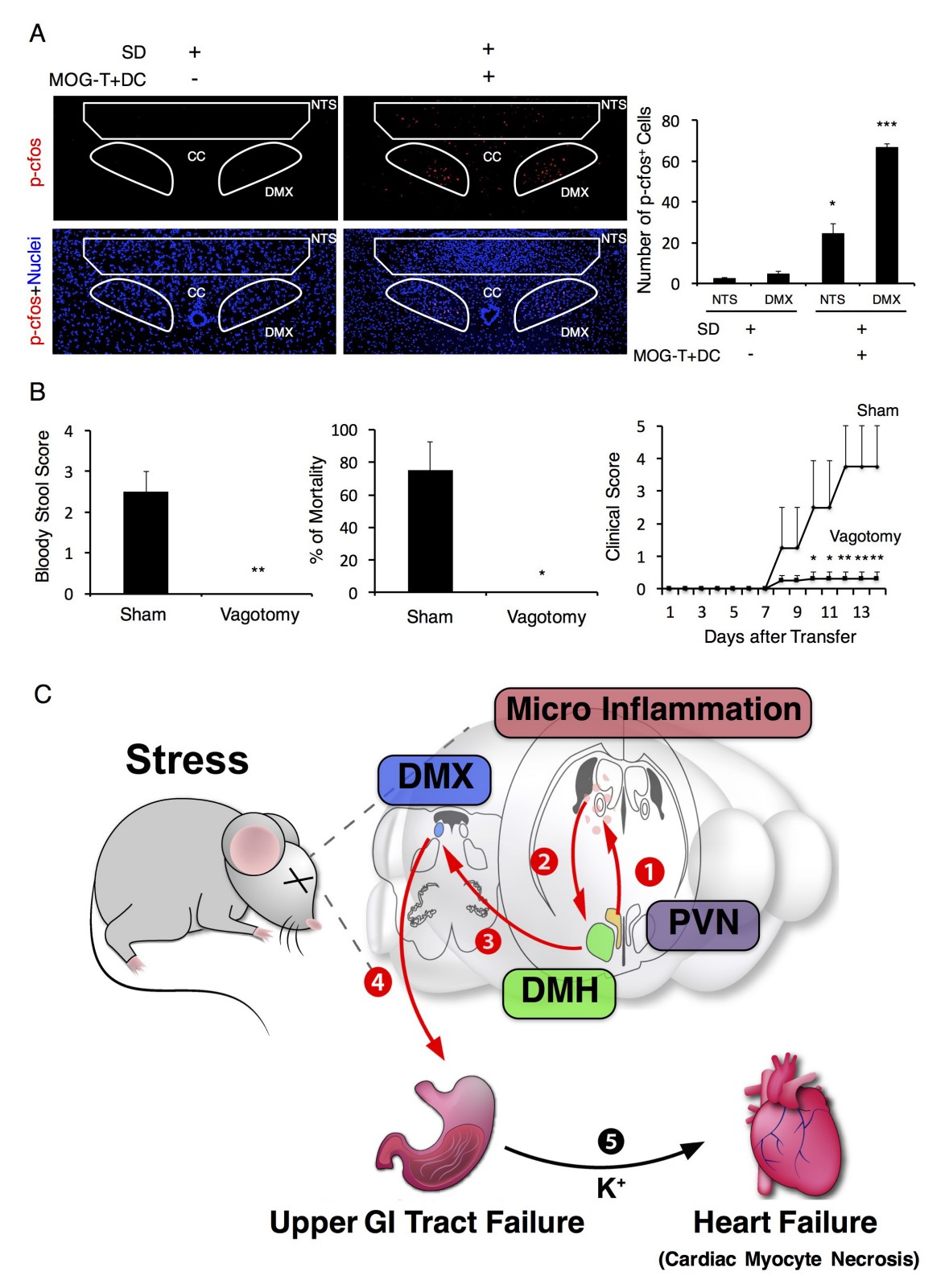

**Figure 7.** Vagal nerve activation induced by brain micro-inflammation under stress condition is critical for the development of severe gastrointestinal failure. (A) Immunohistochemical staining for phospho-cfos in the DMX and NTS of mice under stress condition 2 days after the microinjection of MOG-T plus DC at specific vessels of the boundary area of the third ventricle region, thalamus, and dentate gyrus (n = 3 per group). (B) Bloody stool scores and percentages of mortality of mice under stress condition 2 days after microinjection of MOG-T plus DC at the specific vessels (n = 3–5 per

*Figure 7 continued on next page*

*Figure 7 continued*

group). Clinical scores are also shown (n = 4–5 per group). (**C**) Schematic figure of stress mediated brain micro-inflammation exacerbates gastrointestinal failure and heart failure. (1)The stress mediated PVN activation induced the micro-inflammation at the specific vessels of the boundary area of the third ventricle region, thalamus, and dentate gyrus, followed by (2 and 3) activation of the neurons in the DMH, DMX region and (4 and 5) finally exacerbated gastrointestinal failure and heart failure with cardiac myocyte necrosis. Mean scores ± SEM are shown. Statistical significance was determined by ANOVA tests. Statistical significance is denoted by asterisks (*p<0.05, **p<0.01, ***p<0.001). Experiments were performed at least three times; representative data are shown.

The following figure supplement is available for figure 7:

**Figure supplement 1.** PHA-L-FITC after injection at the DMH reached the DMX.

## Discussion

We here showed that brain micro-inflammation at the boundary area of the third ventricle region, thalamus, and dentate gyrus develops severe gastrointestinal failure via the DMH-vagal pathway in a transfer EAE model under stress. We have reported that pathogenic CD4+ T cells transverse toward the central nervous system (CNS) through a gateway at the dorsal vessels of the fifth lumbar (L5) spinal cord in steady state in response to gravity-mediated chemokine expression via regional sympathetic activation, a phenomenon we call the gravity-gateway reflex (*Arima et al., 2012*). During the gravity-gateway reflex, we found that artificial electric stimulations in different muscles, which mimic specific sensory-sympathetic crosstalk, establsihes the immune cell gateways at different regions of the dorsal vessels of spinal cord, a phenomenon we call the electric-gateway reflex (*Arima et al., 2012*). We also found that pain induces sympathetic-mediated alterations in ventral vessels of the spinal cord to establish gateways to cause EAE relapse, a phenomenon we call the pain-gateway reflex (*Arima et al., 2015*). In the present study, we show that, under stress condition, a gateway forms at the vessels of boundary area of third ventricle region, thalamus, and dentate gyrus, which we describe as the stress-gateway reflex.

We found immune cell accumulation at the specific vessels of brain but not in L5 cord in mice after pathogenic CD4+ T cell transfer under stress (*Figure 3A and B*), although we previously showed that the L5-gateway is established by specific neural activation via gravity-mediated soleus stimulation (*Arima et al., 2012*). It is important to know how was the gateway of immune cells at the L5 cord disappeared? Because we found that mice with stress show reduced movement and often lie down, we hypothesize they would have reduced stimulation of the soleus muscles, which are the main anti-gravity muscles. The sensory neurons from soleus muscles connect to L5 dorsal root ganglions (DRG), which are activated by anti-gravity responses of the soleus muscles in wild type mice. Consistent with our hypothesis, we found that mice with stress reduce L5 DRG activation and chemokine expression from L5 dorsal vessels (*Figure 3—figure supplement 3*). Thus, it is possible that stress condition reduces anti-gravity-mediated soleus stimulation followed by closing the L5 gateway of immune cells.

In our previous work, we identified CX3CL1 as a key chemokine for the accumulation of activated monocytes at L5 ventral vessels and EAE relapse after pain induction in EAE-recovered mice (*Arima et al., 2015*). In detail, the transfer EAE shows transient clinical symptoms. We called EAE mice that stopped showing clinical symptoms EAE-recovered mice. There is no relapse in EAE-recovered mice under normal condition. On the other hand, over 90% of MS patients experience relapse, and there are reports showing that the occurrence of pain is associated with neurologic symptoms and disease severity in patients with MS (*Ehde et al., 2003*, *2006*; *O'Connor et al., 2008*). We therefore hypothesized that pain sensation might induce MS relapse via a reflex pathway. Indeed, pain sensation induced EAE relapse in EAE-recovered mice. Mechanistic analysis showed that the CNS conditions between wild type and EAE-recovered mice are completely different even though both groups show no clinical symptoms. In the EAE-recovered mice, many MHC class IIhiCX3CR1+CD11b+ cells are found in the CNS particulalry in L5 level. These cells are not activated microglial cells but activated monocytes coming from the periphery. The identification of the periphery as the source of MHC class IIhiCX3CR1+CD11b+ cells was done by two methods: parabiosis and labeling microglial cells with CX3CR1-Tdtomato. Importantly, these activated monocytes in the CNS of EAE-

recovered mice stayed for a long time (over 150 days in some cases) after pathogenic CD4+ T cell transfer and are critical for the development of pain-mediated EAE relapse (*Arima et al., 2015*), eLife). On the other hand, many papers have recently reported resident T cells stay in non-immune tissues for a very long time (over 150 days in some cases) after the initial immune response in the tissues and affect subsequent immune responses (*Iijima and Iwasaki, 2014*; *Mueller and Mackay, 2016*). The characteristics of these resident T cells and our activated monocytes in the CNS of the EAE-recovered mice are quite similar: (1) infiltrate the initially affected non-immune tissues, (2) infiltrate a long time, and (3) influence subsequent immune responses. Therefore, we describe the activated monocytes in the CNS of EAE-recovered mice as resident monocytes.

Resident monocytes with CX3CR1 in EAE-recovered mice accumulate from the periphery to stay in the CNS for a long time and contribute to the development of pain-mediated EAE-relapse. Therefore, it is possible that the origin of the resident monocytes could be patrolling monocytes with CX3CR1 in blood stream. We previously showed that resident monocytes with CX3CR1 express CX3CL1 after stimulation with norepinephrine, which is produced around the ventral vessels of the spinal cord following activation of the pain-specific neural pathway (*Arima et al., 2015*). These results suggested a CX3CR1-CX3CL1 autocrine loop in the resident monocytes that is regulated by norepinephrine. A major distinction between the previous and the current study is that in the previous study the activated monocytes were already presented in the CNS as CNS-resident activated monocytes with autoantigen presentation abilityand high CX3CR1 expression in EAE-recovered mice. The CNS-resident activated monocytes were infiltrated the CNS during the initial phase of EAE. The pain-specific neural pathway then induced the accumulation of these CNS-resident activated monocytes at the L5 ventral vessels, which is where norepinephrine was produced via the pain-specific sympathetic pathway, followed by EAE relapse and the accumulation of other immune cells including pathogenic CD4+ T cells from the blood stream. However, in the current study, there are no CNS-resident activated monocytes in wild type mice under stress. Therefore, CX3CL1, which is a critical chemokine for CNS-resident monocytes, is not critical for the stress-gateway reflex in this study. On the other hand, during the stress-gateway reflex, we recognized CCL5 is required for the recruitment of pathogenic CD4+ T cells, including Th1 and Th17 cells, to the specific vessels from the periphery to trigger micro-inflammation and stimulate the subsequent recruitment of MHC class IIhiCD11b+ cells. (These MHC class IIhiCD11b+ cells are possibly similar or the same cell types as the activated monocytes in the pain-gateway reflex, although majority of activated monocytes from the periphery initially express low CX3CR1 in the CNS). We assume that the recruitment of MHC class IIhiCD11b+ cells may be at least partially dependent on CX3CL1 from endotheial cells, which is induced by cytokines from pathogenic CD4+ T cells that accumulated at the specific vessels in response to CCL5. The cytokines IFN-γ, IL-17A, and IL-6, which are likely secreted by the pathogenic CD4+ T cells, are also important for the development of this micro-inflammation and the resulting severe gastrointestinal failure. Indeed, both the accumulation of CD4+ T cells and MHC class IIhiCD11b+ cells at the specific vessels as well as the severe gastrointestinal failure were suppressed by blockade of CCL5 or IL-17A and IFN-γ. Additionally, IL-17A- or IFN-γ-deficient pathogenic CD4+ T cells did not accumulate at the specific vessels even under stress. We hypothesized the BBB breaching at the specific vessels depends on antigen-presenting events on pathogenic CD4+ T cells. In agreement, we found that MHC class IIhiCD11b+ cells had the ability to activate pathogenic CD4 + T cells without exogenous MOG peptide (*Figure 4—figure supplement 5*), suggesting that MHC class IIhiCD11b+ cells have autoantigen-presentation ability. We theorize that the chemokine/cytokine expression and the accumulation of MHC class IIhiCD11b+ cells as well as pathogenic CD4+ T cells at specific vessels could be used to diagnose and/or therapeutically target several types of severe gastrointestinal failure in patients.

We found that the development of fatal gastrointestinal failure depended on activation of a previously unidentified DMH-vagal nerve pathway, which was triggered by ATP from the brain micro-inflammation sites. We here found that this nerve pathway was hardly activated under steady state or stress conditions without brain micro-inflammation. Thus, ATP at the specific vessels can be viewed as a switch that activates this new neural pathway. At the same time, the brain micro-inflammation is triggered by PVN-mediated TH+ sympathetic activation in the presence of pathogenic CD4+ T cells in the blood. Emotional responses can activate specific nuclei in emotional centers that contain TH+ noradrenergic neurons. We found TH+ noradrenergic neurons connected the PVN with the specific vessels localized at the boundary area of the third ventricle region, thalamus, and

dentate gyrus. There exist TH+ noradrenergic neuronal pathways that connect other nuclei to some specific vessels in the brain. For example, it is reported that cerebral microvessels have noradrenergic innervation from the locus coeruleus (*Harik, 1986*; *Kalaria et al., 1989*). Our results therefore may indicate that brain micro-inflammation at some specific vessels could activate unidentified neural pathways to regulate the homeostasis of various organs including brain itself.

The brain-gut axis involves several neural components, including (1) the autonomic nervous system, (2) the CNS, (3) the stress system such as the hypothalamic-pituitary-adrenal axis, and (4) the corticotropin-releasing factor system, along with the intestinal response (*Caso et al., 2008*). Regarding connection between (1) the autonomic nervous system and (2) the CNS, we showed that the stress condition activates PVN noradrenagic/sympathetic neurons connected to vessels localized at the boundary area of the third ventricle region, thalamus, and dentate gyrus. This activation is important for the accumulation of pathogenic CD4+ T cells and MHC class IIhiCD11b+ cells. Regarding connection between (2) the CNS and (3) the stress system, the enhanced activation of the neurons in the PVN/DMH regions by the brain micro-inflammation and stress activated DMX/vagal nerves (*Figure 7C*). During this process, (4) the corticotropin-releasing factor system was also activated, as indicated by the high corticosteroid levels in the serum of mice even without pathogenic CD4+ T cell transfer. Importantly, we found that gastrointestinal diseases after microinjection of cytokines in mice with stress were not suppressed by two antagonists of the corticosteroid receptor such as mifepristone and guggulsterone (*Figure 1—figure supplement 3*). Thus, regarding the gastrointestinal response, the resulting vagal nerve activation induced severe gastrointestinal failure, although the hypothalamic-pituitary-adrenal axis has a minimum role in the development of micro inflammation-mediated gastrointestinal diseases under stress.

How is the linkage between pathogenic CD4+ T cells and activation of p38 in stomach? It is known that pathogenic CD4+ T cells for the transfer EAE express cytokines including IFN-γ, IL-17A, and IL-6 (*Langrish et al., 2005*), and very strong stress alone induces gastric disease via p38 activation at the affected tissues by the activation of the vagal pathway (*Debas and Carvajal, 1994*; *Jia et al., 2007*; *Uwada et al., 2017*). We showed that the injection of IFN-γ, IL-17A, and IL-6 or ATP at specific vessels in the brain of mice with relatively low stress establishes a neural pathway via DMH and DMX (nucleus of vagal nerves) followed by the development of fatal gastrointestinal diseases (*Figures 4G* and *5G*) and the activation of p38 in stomach (*Figure 2—figure supplement 1*). Moreover, cytokine stimulation in endothelial cells causes the expression of ATP (*Figure 6A*). These results strongly suggested that cytokines from pathogenic CD4+ T cells act on endothelial cells at the specific vessels and induce ATP around the specific vessels. The resulting ATP activates the DMH-vagal axis, which is important for the development of gastrointestinal diseases as well as the activation of p38. Therefore, cytokines from the pathogenic CD4+ T cells are a triggering factor to activate p38 in the affected GI tract tissues via DMH-vagal axis activation in mice with relatively low-stress.

Several papers have reported concurrence of MS and inflammatory bowel diseases (IBD) (*Gupta et al., 2005*; *Kimura et al., 2000*; *Pokorny et al., 2007*; *Rang et al., 1982*; *Sadovnick et al., 1989*). Interestingly, primary progressive MS is dominant in patients who suffered from bowel dysfunction (*Preziosi et al., 2013*), suggesting that transfer EAE under stress condition could model progressive MS. Complications of celiac disease, epilepsy and cerebral calcifications are also known to be associated with MS (*Gobbi, 2005*; *Gobbi et al., 1992*). Therefore, our model may provide a possible explanation for several brain-intestine comorbidities during brain diseases including MS.

It is reported that micro-inflammations in the brain occur in patients with neurodegenerative diseases such as Alzheimer's disease, non-Alzheimer type dementia and Parkinson's disease (*Appel et al., 2010*; *Togo et al., 2002*), psychological disorder (*Najjar et al., 2013*), and epilepsy (*Vezzani et al., 2011*). Cerebral microbleeding is an important risk factor for dementia (*Watanabe et al., 2016*). Moreover, some of these diseases are known to be associated with specific allies of MHC class II genes (*Ferrari et al., 2014*; *Nalls et al., 2014*), suggesting involvement of autoreactive CD4+ T cells. Based on our findings, we hypothesize that micro-inflammation at some vessels, which is most likely triggerred by autoreactive CD4+ T cells, could activate new neural pathways to cause dysfunction in the brain neural network and/or in certain organs in the periphery. It is reported that the prevalence of dementia is considerably higher in elder people with gastritis and

psychological disorders is associated with irritable bowel syndrome (IBS) (*Momtaz et al., 2014*), which supports our hypothesis.

Sensitivity to chronic stress in humans is diverse owing to genetic factors and/or environmental factors such as stressful events in early life when the brain is particularly sensitive to stress (*Lupien et al., 2009*; *Mayer, 2000*). Indeed, it is reported that early life stress such as abuse or poor care plays a role in the susceptibility to develop behavior problems and gastrointestinal diseases (*Lupien et al., 2009*; *Mayer, 2000*). The early life period is also important for the establishment of T cell repertoire and central tolerance, and it is known that thymocytes are very sensitive to glucocorticoid that can be induced by stress (*Ashwell et al., 2000*; *Hogquist et al., 2005*). Therefore, it is tempting to speculate that individual variations to respond to stress might also be defined by the presence of autoreactive CD4+ T cells to the CNS autoantigens, and that measurement of the autoreactive CD4+ T cell population in the peripheral blood might predict the susceptibility to develop peripheral organ failure by chronic stress. Genetic associations of certain MHC class II alleles with diseases including dementia and neurodegenerative diseases etc that involve brain micro-inflammation described above support our hypothesis (*Ferrari et al., 2014*; *Nalls et al., 2014*).

In summary, we demonstrated a molecular mechanism of the brain-gut axis by using a brain auto-immune-inflammation model under stress condition. Our findings provide molecular insight on how brain micro-inflammation at specific vessels develops fatal gastrointestinal failure. Because there are afferent and efferent neural systems in whole body, it could be possible that dysregulations in the peripheral organs including not only gastrointestine and heart but also other organs might also affect the status or region of micro-inflammation in the brain. We propose that local brain micro-inflammation is induced by stress-mediated PVN activation to enhance the activation of a new neural pathway, the DMH-vagal nerve pathway, to aggravate gastrointestinal failure.

## Materials and methods

### Mouse strains

C57BL/6 mice were purchased from Japan SLC (Tokyo, Japan). C57BL/6-PL mice were purchased from Taconic (Germantown, NY). In addition, C57BL/6-Tg (Tcra2D2,Tcrb2D2)1Kuch/J (*Bettelli et al., 2003*), IL-17A-deficient mice (*Nakae et al., 2002*) and B6.129S7-Ifng$^{tm1Ts}$/J (*Dalton et al., 1993*) were used in this study. There are no sample exclusion criteria. Sample size of more than three mice was chosen to ensure power for statistical tests unless the availability of mice was limited. All mice were maintained under specific pathogen-free conditions according to the protocols of Hokkaido University. The animal experiments used in this study were approved by the Institutional Animal Care and Use Committees of Hokkaido University (Approval number: 14–0083).

### Passive transfer of pathogenic CD4+ T cells from mice to induce EAE

EAE induction was performed as described previously (*Arima et al., 2012*, *2015*; *Ogura et al., 2008*). Briefly, C57BL/6 mice were injected with a MOG(35-55) peptide (Sigma-Aldrich, Tokyo) in complete Freund's adjuvant (Sigma-Aldrich) at the base of the tail on day 0 followed by intravenous injection of pertussis toxin (Sigma-Aldrich) on days 0, 2, and 7. On day 9, CD4$^+$ T cells from the resulting mice were sorted using anti-CD4 microbeads (Miltenyi Biotec, Tokyo). The resulting CD4$^+$ T cell-enriched population ($4 \times 10^6$ cells) was cocultured with rIL-23 (10 ng/ml; R&D Systems, Minneapolis, MN) in the presence of MOG peptide-pulsed irradiated splenocytes ($1 \times 10^7$ cells) for 2 days. Anti-CD4 microbeads were used to enrich CD4+ T cells. These pathogenic CD4+ T cells ($1.5 \times 10^7$ cells) were then injected intravenously into wild type mice. Clinical scores were measured as described previously (*Arima et al., 2012*, *2015*; *Ogura et al., 2008*). As examples of non-CNS antigens, OVA(323-339) peptide (Sigma-Aldrich, Tokyo) and human IRBP (1-20) peptide (Sigma-Aldrich, Tokyo) were used. Except for the peptides, OVA- and IRBP-specific CD4+ T cells were generated and transferred in the same way as MOG-pathogenic CD4+ T cells.

### Fecal occult blood test

Stool samples were collected from each mouse and dissolved in saline. After centrifugation (8000 rpm, 5 min), the supernatant was collected and diluted tenfold with saline. The fecal occult blood test was performed with hemastix (SIEMENS, Germany).

## Psychophysiological stress-induced sleep disorder model

6–8 week-old C57BL/6 mice were individually maintained in plastic cages with running wheels for habituation. After that, the mice were exposed to psychophysiological stress to induce sleep disorder (SD) (*Miyazaki et al., 2013*; *Oishi et al., 2014*). Briefly, paper-chip bedding was replaced with water to a depth of 1.5 cm, which caused the mice to run on the wheel all day. After 2 days of SD, the mice received pathogenic CD4+ T cells intravenously.

## Wet bedding stress model

6–8 week-old C57BL/6 mice were put in damp bedding (350 ml water in a cage), which was changed daily. After 2 days of stress treatment, the mice were injected i.v. with pathogenic CD4+ T cells.

## Measurement of potassium levels

Potassium levels in plasma were determined using Fuji Dri-Chem 7000 (FUJIFILM, Japan).

## ATP assay

A type 1 collagen+ endothelial BC1 cell line was obtained from Dr. M. Miyasaka (Osaka University). For stimulation, BC1 cells were plated in 96-well plates ($1 \times 10^4$ cells/well) and stimulated with human IL-6 (50 ng/ml; Toray Industries) plus human soluble IL-6 receptor $\alpha$ (50 ng/ml; R&D Systems), mouse IL-17A (50 ng/ml; R&D Systems). ATP was determined with a luciferin-luciferase assay using an ATP assay kit (TOYO Ink, Tokyo) according to the manufacturer's instructions.

## Multiplex assay

Cortisol levels in serum were determined using a milliplex kit from Merck Millipore (Tokyo)

## Measurement of aldosterone levels

Aldosterone levels in serum were determined using a ELISA kit from ENDOCRINE (USA).

Western Blotting

Gastric mucosa samples were homogenized in protein lysis/sample buffer (RIPA buffer: 50 mM Tris, pH8.0, 0.1% sodium dodecyl sulfate, 1% Nonidet P-40, 0.5% deoxycholate, 150 mM NaCl, 1 mM PMSF) containing a protease inhibitor and proteasome inhibitor and centrifuged (15,000 rpm, 20 min), and the resultant supernatants used as protein lysates. The concentration of protein was measured with protein assay kit (Bio-rad, USA) followed by SDS-PAGE (Wako, Tokyo, Japan). After transfer to a polyvinylidene difluoride membrane, immunoblotting was performed according to the manufacturer's protocol (Cell Signaling Technology).

## Cardiac electrocardiogram system

A telemetry transmitter (TA11ETA-F10; Data Science International, USA/Primetech Corporation, Tokyo) was implanted subcutaneously into the flank of the body. Electrocardiograph (ECG) electrodes were fixed on both side of the chest. The monitoring was performed according to the manufacturer's protocol by using Dataquest A,R,T Acquisition (Primetech Corporation, Tokyo). Analysis was performed with Dataquest A,R,T Analysis (Primetech Corporation, Tokyo).

## Measurement of hematocrit levels

Blood was collected in heparinized microhematocrit capillary tubes and centrifuged for 5 min at 12,000 rpm. After that, the percentage of hematocrit was determined with a hematocrit reader from KUBOTA (Tokyo).

## Histological analysis

Brain was harvested, embedded in SCEM compound (SECTION-LAB Co. Ltd., Hiroshima, Japan), and prepared as sections using the microtome device CM3050 (Leica Microsystems, Tokyo) with Cryofilm type IIIC (16UF) from SECTION-LAB Co. Ltd (Tokyo). The resulting sections were stained with hematoxylin/eosin or immunohistochemical staining and analyzed with a BZ-9000 microscope (KEYENCE, Osaka, Japan). Analysis was performed by HS ALL software in BZ-II analyzer (KEYENCE). Frozen sections (10 μm) were prepared according to a published method (*Arima et al., 2012*; *Kawamoto, 2003*).

## Antibodies and reagents

The following antibodies were used for the flow cytometry analysis: FITC-conjugated anti-CD19 (RRID:AB_464966, eBioscience, Tokyo), anti-CD11b (RRID:AB_312788, BioLegend, Tokyo), anti-CD44 (RRID:AB_493685, BioLegend), anti-CD4 (RRID:AB_312712, BioLegend), anti-NK1.1 (RRID:AB_465317, eBioscience), PE-conjugated anti-CD44 (RRID:AB_312959, BioLegend), anti-TCR$\beta$(RRID:AB_313430, BioLegend), anti-CD11b (RRID:AB_312791, BioLegend), PE-Cy7-conjugated anti-CD90.2 (RRID:AB_469642, eBioscience), anti-CD8 (RRID:AB_312760, BioLegend), PerCP-conjugated anti CD3 (RRID:AB_893319, BioLegend), APC-conjugated anti-CD4 (RRID:AB_312718, BioLegend, Tokyo), anti-I-A/I-E (RRID:AB_313329, BioLegend), biotin-conjugated anti-CD11b (RRID:AB_312787, BioLegend), anti-CD19 (RRID:AB_312823, eBioscience), anti-NK1.1 (RRID:AB_466804, eBioscience), anti-CD11c (RRID:AB_313772, BioLegend), anti-TCR$\alpha$ (RRID:AB_313426, BioLegend), and Pacific Blue-conjugated anti-Gr-1 (RRID:AB_893559, BioLegend), anti-$\gamma\delta$TCR (RRID:AB_466669, eBioscience). The following antibodies were used for immunohistochemistry: FITC-conjugated anti-I-A/I-E (RRID:AB_313321, BioLegend), anti-Dopamine Transporter (RRID:AB_305226, Abcam, Tokyo), anti-Noradrenaline Transporter (RRID:AB_305477, Abcam), anti-phospho-CREB (RRID:AB_2561044, Cell Signaling Technology, Tokyo), anti-tyrosine hydroxylase (RRID:AB_1524535, Abcam), anti-Phospho-c-Fos (Ser32) (RRID:AB_10557109, Cell Signaling Technology), control rabbit IgG (DA1E) (RRID:AB_1550038, Cell Signaling Technology), biotin-conjugated anti-CD4 (RRID:AB_312710, BioLegend), anti-CD11b (RRID:AB_312787, BioLegend), anti-I-A/I-E (RRID:AB_313319, BioLegend), anti-P2 $\times$ 7 receptor (RRID:AB_881835, Abcam), anti-NFATc1 (RRID:AB_2152503, Santa cruz), anti-Neurofilament L (RRID:AB_10828120, Cell Signaling Technology), anti-Phaseolus vulgaris Agglutinin (RRID: AB_10000080, VECTOR, Burkingmem, CA), Alexa Fluor 546 donkey anti-goat IgG (H + L) (RRID:AB_142628), Alexa Fluor 488 goat anti-rabbit IgG (H + L) (RRID:AB_2576217), Alexa Fluor 546 goat anti-rabbit IgG (H + L)(RRID:AB_143051), Alexa Fluor 647 goat anti-rabbit IgG (H + L)(RRID:AB_141775), Alexa Fluor 647 goat anti-chicken IgG (RRID:AB_1500594) (Invitrogen, Tokyo), and Streptavidin Alexa Fluor 546 conjugate (RRID:AB_2532130, Invitrogen). The following antibodies were used for western blotting: mouse anti-tubulin antibody (RRID:AB_477579, Sigma-Aldrich), rabbit anti-p38 antibody (RRID:AB_10998134, Cell Signaling Technology), rabbit anti-phospho p38 (Thr180/Tyr182) antibody (RRID:AB_2139682, Cell Signaling Technology), rabbit anti-MAPKAPK-2 antibody (RRID:AB_2235082, Cell Signaling Technology), anti-phospho MAPKAPK 2 (Thr334) antibody (RRID:AB_490938, Cell Signaling Technology). The following antibodies were used for in vivo neutralization: anti-mouse IL-17 Ab (RRID:AB_2125018), anti-CCL5 Ab (RRID:AB_355385), anti-CCL2 Ab (RRID:AB_354500), anti-CX3CL1 Ab (RRID:AB_2276839) (R&D Systems). anti-IFN-$\gamma$ antibody was purified as described previously (*Ueda et al., 2006*). 6-Hydroxydopamin hydrochloride, Lansoprazol, FITC-CTB, Tamoxifen, Muscimol, ATP, Mifepristone and guggulsterone were purchased from Sigma-Aldrich. PHA-L was purchased from VECTOR. A438079 was purchased from TOCRIS Bioscience (Minneapolis, MN).

## Flow cytometry

To generate single cell suspensions, brains or retina were dissected after cardiac perfusion and enzymatically digested using the Neural Tissue Dissection Kit (Miltenyi Biotec) or collagenase D (Basel, Switzerland), and $10^6$ cells were incubated with fluorescence-conjugated antibodies for 30 min on ice for cell surface labeling. The cells were then analyzed with cyan flow cytometers (Beckman Coulter, Tokyo). The collected data were analyzed using Summit software (Beckman Coulter) and/or Flowjo software (Tree Star, Ashland, OR).

## Immunohistochemistry

Immunohistochemistry was performed as described previously with slight modifications (*Lee et al., 2012*).

## Laser micro-dissection

Approximately 100 frozen sections (15 μm/section) were fixed with PAXgene (QIAGEN, Tokyo) for 15 min followed by 100% EtOH for 10 min. Tissues around the third ventricle vessels in the sections were collected by a laser micro-dissection device, DM6000B (Leica Microsystems), and total RNA

was extracted by the RNeasy micro Kit (QIAGEN). After DNase treatment and reverse transcription, cDNA was subjected to real-time qPCR analysis.

## Real-time PCRs

The GeneAmp 5700 sequence detection system (ABI, Tokyo) and KAPA PROBE FAST ABI Prism qPCR Kit (Kapa Biosystems, Boston, MA) were used to quantify the levels of HPRT mRNA and CCL5 mRNA. The PCR primer pairs used for real-time PCRs with the KAPA PROBE FAST ABI Prism qPCR Kit were as follows: mouse *Hprt* primers, 5'-AGCCCCAAAATGGTTAAGGTTG −3' and 5'-CAAGGGCATATCCAACAACAAAC-3', probe, 5'-ATCCAACAAAGTCTGGCCTGTATCCAACAC-3'; mouse *Ccl5* primers, 5'-CTCCCTGCTGCTTTGCCTAC-3' and 5'-CGGTTCCTTCGAGTGACAAACA-3', probe, 5'-TGCCTCGTGCCCACGTCAAGGAGTATT-3'; mouse *Ccl20* primers, 5'-ACGAAGAAAA-GAAAATCTGTGTGC-3' and 5'-TCTTCTTGACTCTTAGGCTGAGG-3', probe, 5'-AGCCCTTTTCACC-CAGTTCTGCTTTGGA-3'; mouse *cfos* primers 5'-CCTTCTCCAGCATGGGCTC-3' and 5'-CG TGGGGGATAAAGTTGGCACTA-3', probe, 5'-TGTCAACACACAGGACTTTTGCGCAGAT-3'. The conditions for real-time PCR were 40 cycles at 95°C for 3 s followed by 40 cycles at 60°C for 30 s. The relative mRNA expression levels were normalized to the levels of HPRT mRNA.

## Treatments of antibodies and reagents

In some experiments, anti-IFN-γ antibody (100 μg/mouse), anti-IL-17A antibody (100 μg/mouse), anti-CCL2 antibody (100 μg/mouse), anti-CX3CL1 antibody (100 μg/mouse) or anti-CCL5 antibody (100 μg/mouse) were intraperitoneally injected everyday after pathogenic CD4+ T cell transfer. Lansoprazol (30 mg/kg) was treated orally everyday after pathogenic CD4+ T cell transfer.

## Brain microinjection

The head of an anesthetized mouse was placed in a stereotaxic device. Fur above the skull was shaved, and the skin was cleaned with 70% ethanol. A 30-gauge needle was lowered toward the third ventricle vessels (AP −1.06 mm; ML 1 mm; DV 2.25 mm), PVN (AP −1.06 mm; ML 0.25 mm; DV 4.8 mm), and DMH (AP −1.46 mm; ML 0.37 mm; DV 5 mm), and 6-OHDA, FITC-CTB, PHA-L, Muscimol (2 mg/ml, 1 mg/ml, 25 mg/ml, 0.25 mg/ml, respectively, 0.2 μl each delivered over 90 s) were injected as described previously (*Kim et al., 2011*). Pathogenic CD4+ T cells (1 × 10$^6$ cells) plus MOG-pulsed BMDC (5 × 10$^5$ cells) were injected around the third ventricle vessels by the same protocol. IL-6 (50 ng; Toray) + IL-17A (50 ng; R&D Systems), IFN-γ (50 ng; PeproTech, Tokyo) + IL-17A (50 ng), γATP (2 μg), and A438079 (1 μg) were injected around the third ventricle vessels by the same protocol. Mifepristone (30 mg/kg; Sigma), guggulsterone (30 mg/kg; Sigma) were intraperitoneally injected everyday after cytokine injection.

## Surgical procedures

Anesthetized mice were placed in a stereotactic frame, and a hole was drilled through the skull. An electrode (Brain Science Idea, Tokyo) was inserted through the skull at the level of the PVN (AP −1.06 mm; ML 0.25 mm; DV 4.8 mm), and a direct current of 400 uA was applied for 5 s.

## Subdiaphragmatic vagotomy

The stomach and lower esophagus were visualized from an upper midline laparotomy. The stomach was gently retracted down beneath the diaphragm to clearly expose both vagal trunks. At least 1 mm of visible vagal nerve was dissected. In addition, all neural and connective tissue surrounding the esophagus immediately below the diaphragm was removed to transect all small vagal branches.

## Immobilization stress

EAE-recovered mice were subjected to immobilization stress in a plastic tube for 30 min/day over 2 days (*Yoshihara and Yawaka, 2013*).

## Antigen presentation assay

Naïve CD4+ T cells from 2D2 mice and CD11b$^+$ cells from SD+T cells+ mice were sorted using a cell sorter (MoFlo, Beckman) and anti-CD11b microbeads, respectively (Miltenyi Biotec). The resulting CD4$^+$ T cell-enriched population (1 × 10$^5$ cells) was cocultured with the isolated CD11b$^+$ cells

(5 × 10$^4$ cells or 1 × 10$^5$ cells) without MOG-peptide addition in a 96 well plate for 2 days. IL-2 levels in cell culture supernatants were determined using ELISA kits (eBioscience).

## Statistical analysis

Student's t tests (two-tailed) and ANOVA tests were used for the statistical analysis of differences between two groups and that of differences between more than two groups, respectively. P values less than 0.05 were considered to be statistically significant.

## Acknowledgements

We appreciate the excellent technical assistances provided by Ms. Ezawa, and Ms. Nakayama, and thank Ms. Fukumoto for her excellent assistance. We thank Dr. P Karagiannis (CiRA, Kyoto University, Kyoto, Japan), Dr. T Hirano (QST, Chiba, Japan), Dr. K Tainaka (Niigata University) for carefully reading the manuscript and/or important discussion, respectively. This work was supported by KAKENHI (D K, Y A, and M M), Takeda Science Foundation (M M), Institute for Fermentation Osaka (M M), Mitsubishi Foundation (M M), Mochida Memorial Foundation for Medical and Pharmaceutical Research (D K), Suzuken Memorial Foundation (Y A), Japan Prize Foundation (Y A), Ono Medical Research Foundation (Y A), Kanzawa Medical Research Foundation (Y A), Kishimoto Foundation (Y A), Nagao Takeshi Research Foundation (Y A), Japan Multiple Sclerosis Society (Y A), Kanae Fundation (Y A), The Uehara Memorial Fundation (Y A), Japan Brain Fundation (Y A), The Kao Fundation for Arts and Sciences (Y A), Tokyo Medical Research Foundation (M M and Y A), JSPS Postdoctoral Fellowship for Foreign Researchers (A S), and the Joint Usage/Research Promotion Office at Institute for Genetic Medicine, Hokkaido University (M M). We dedicate this manuscript to Drs. H Nakayama, R Fukuhara, and Y Sato (Hyogo Prefectural Amagasaki General Medical Center), who saved Dr. Murakami's life from angina pectoris during its preparation.

## Additional information

### Funding

| Funder | Grant reference number | Author |
|---|---|---|
| Japan Society for the Promotion of Science | KAKENHI 15K19122 | Yasunobu Arima |
| Takeda Science Foundation | | Masaaki Murakami |
| Institute for Fermentation, Osaka | | Masaaki Murakami |
| Mitsubishi Foundation | | Masaaki Murakami |
| Mochida Memorial Foundation for Medical and Pharmaceutical Research | | Daisuke Kamimura |
| Suzuken Memorial Foundation | | Yasunobu Arima |
| Japan Prize Foundation | | Yasunobu Arima |
| Kishimoto Foundation | | Yasunobu Arima |
| Nagao Takeshi Research Foundation | | Yasunobu Arima |
| Japan Multiple Sclerosis Society | | Yasunobu Arima |
| Kanae Foundation for the Promotion of Medical Science | | Yasunobu Arima |
| Tokyo Medical Research Foundation | | Yasunobu Arima Masaaki Murakami |
| Japan Society for the Promotion of Science | Postdoctal Fellowship for Foreign Researchers | Andrea Stofkova |
| Ono Medical Research Foun- | | Yasunobu Arima |

dation

| | | |
|---|---|---|
| Kanzawa Medical Reseach Foundation | | Yasunobu Arima |
| Uehara Memorial Foundation | | Yasunobu Arima |
| Japan Brain Foundation | | Yasunobu Arima |
| Kao Foundation for Arts and Sciences | | Yasunobu Arima |
| Japan Society for the Promotion of Science | KAKENHI 15K08518 | Daisuke Kamimura |
| Japan Society for the Promotion of Science | KAKENHI 15H04741 | Masaaki Murakami |

The funders had no role in study design, data collection and interpretation, or the decision to submit the work for publication.

## Author contributions

YA, Data curation, Formal analysis, Funding acquisition, Validation, Investigation, Writing—original draft; TO, NN, Data curation, Formal analysis, Validation, Investigation, Writing—original draft; KH, Data curation, Validation; MO, Data curation, Formal analysis, Investigation; YT, ME, Data curation, Formal analysis, Validation, Investigation; JN-K, YMor, TK, MK, TI, YY, MT, SS, Methodology; RS, NS, AY, MP, Resources; AS, Data curation, Validation, Formal analysis, Invesitgation; YS, Data curation, Investigation; YMor, Supervision, Investigation, Project administration; DK, Formal analysis, Funding acquisition, Investigation, Writing—original draft, Writing—review and editing; MM, Conceptualization, Supervision, Funding acquisition, Investigation, Writing—original draft, Project administration, Writing—review and editing

## Author ORCIDs

Andrea Stofkova, http://orcid.org/0000-0002-6579-6578
Masaaki Murakami, http://orcid.org/0000-0001-7159-7279

## Ethics

Animal experimentation: All mice were maintained under specific pathogen-free conditions according to the protocols of Hokkaido University. The animal experiments used in this study were approved by the Institutional Animal Care (Approval number : 14-0083).

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
