## [Decision Letter]

Thank you for submitting your article "Regional brain inflammation at specific vessels exacerbates gastrointestinal failure under stress conditions" for consideration by *eLife*. Your article has been favorably evaluated by Tadatsugu Taniguchi (Senior Editor) and three reviewers, one of whom is a member of our Board of Reviewing Editors. The reviewers have opted to remain anonymous.

The reviewers have discussed the reviews with one another and the Reviewing Editor has drafted this decision to help you prepare a revised submission.

Summary:

The authors demonstrate that stress conditions increase clinical signs of EAE in a transfer model, including increased migration of CD4 T cells to ventricle areas of the brain. Stressed mice further developed severe GI conditions, specifically within the stomach and duodenum, with the suggestion a dysregulation of secretion or reflux of bile acid (or other acids) within the stomach. Mice also succumbed to myocardial infarction, potentially due to increases in potassium levels within circulation that followed the severe cellular destruction within the stomach and intestine. The authors go on to describe a chemokine/receptor circuit involved in the increased recruitment of immune cells to the brain and the nerve cells that transmit the signals that potentiate the stomach/intestinal damage.

Overall, this is a very nice study of crosstalk between the immune system and nervous system that leads to major alterations in physiology and health. However, there are several concerns as detailed below that reduce enthusiasm for the manuscript, and addressing them would be essential.

Essential revisions:

1) It would be useful to show that non-CNS-autoantigen-specific CD4 T cells do not mediate these effects.

2) More supporting evidence would be useful for implicating the HPA axis, which is currently done on the basis of higher blood aldosterone levels in stressed versus unstressed mice.

3) It is not clear that the combination of chronic stress and autoimmune CD4 T cells causes 'atypical EAE', since no data regarding the EAE itself are provided. In fact, the manuscript does not provide any information on whether the L5-specific point of initial autoimmune inflammation they have identified earlier is ALSO operative in chronically stressed recipients leading to an unaltered clinical course of neurological defects, or whether that entry point is no longer used, leading a far more radical restructuring of the clinical syndrome. Further, the finding that lansoprazol treatment completely eliminates clinical signs may be an indication that EAE itself may not be prominent in this complex situation of stress and autoimmunity.

4) While the CNS studies are very thorough, there is little information describing their functional linkage to the stomach epithelial destruction, and about how stress 'primes' the stomach for receiving the nerve signals that initiate the epithelial damage.

5) The manuscript provides no explanation of the relevant molecular pathway distinctions between acute stress versus chronic stress that lead to the latter, but not the former, causing the disease phenotype addressed.

6) The manuscript provides no insights into the relevant distinctions that make CX3CL1 the primary neuro-immune gateway chemokine at L5 in earlier work from the authors, and CCL5 the gateway chemokine in the present data.

7) The manuscript does not address whether the IFNgamma and IL-17 cytokines, relevant for the pathway shown, are necessarily made by the pathogenic autoreactive CD4 T cells.

8) The manuscript does not show whether the role of local CCL5 is merely as a gateway for inflammation, in which case the pathological consequences of local injection either T cells or cytokines would be expected to be CCL5-independent.

9) The manuscript does not address the cellular provenance of ATP sensing (is the ATP receptor needed on immune lineage cells and/or on neural lineage cells, for example?), and more broadly, it does not provide any positioning of ATP sensing in the pathogenic pathway it identifies (is it upstream or downstream of CCL5, IFNγ/IL17, neuronal activation…).

10) In fact, it would be very useful to have a cartoon-model of the pathway to provide an overview of the connections being made.

11) As a textual point, the manuscript perhaps makes too many MS-specific claims; EAE may not be a sufficiently robust MS model for that. The data are remarkable enough independent of what they may say about MS.

12) It would be very useful to have the 'data not shown' to be shown at least in supplementary figures.

[Editors' note: further revisions were requested prior to acceptance, as described below.]

Thank you for resubmitting your work entitled "Brain micro-inflammation at specific vessels dysregulates organ-homeostasis via the activation of a new neural circuit" for further consideration at *eLife*. Your revised article has been favorably evaluated by Tadatsugu Taniguchi (Senior Editor), a Reviewing Editor and two reviewers.

The manuscript has been improved but there are some remaining issues that need to be addressed, at least by appropriate textual modifications, before acceptance, as outlined below:

The manuscript by Murakami and his group has been substantiated by much additional data, accumulated in an attempt to address many of the key issues raised by the referees.

Yet, some of the added data provide correlative information, rather than mechanistic insight. In addition, the manuscript contains several inaccuracies.

General comments:

The authors speak about pathogenic T cells throughout the manuscript, without even relating to CD8 cells that are very much associated with EAE. Moreover, based on current models, IFN-γ does not have a major role in the pathology.

The authors also describe resident monocytes in the brain. There are no resident monocytes in the brain, under any circumstances. The only resident myeloid cells in the brain are microglia, even under pathology; in EAE, the infiltrating monocytes should be termed monocyte-derived macrophages, and unlike the resident microglia, they do not express CX3CR1+.

Specific comments:

1) The authors were requested to add a control consisting of autoimmune cells with non-CNS specificity – Ova specific T cells are not an appropriate control; the question is whether autoimmune T cells directed against self-antigens in areas outside the brain will affect the gut.

2) The authors were asked to add results relating to the HAPaxis; the additional measurements of cortisol do not help. Such experiments provide a correlation but not causal results. Blocking of the corticosteroid receptor or its function could add to our understanding of the mechanism.

3) No mechanistic explanation is provided for the changes in the autoimmune T cell gateway from L5 to the brain.

4) No explanation is given for the linkage between the T cells and activation of p38.

5) The role of Cx3CL1 expression by monocytes is puzzling; the receptor is normally expressed by patrolling monocytes or within the CNS by microglia.

---

## [Author Response]

*Essential revisions:*

*1) It would be useful to show that non-CNS-autoantigen-specific CD4 T cells do not mediate these effects.*

We have added data of clinical scores and mortality for ovalbumin (OVA)-specific

CD4^+^ T cell transfer to sleep disorder (SD) mice and water-bedding stress (WB) mice. Non-CNS-autoantigen-specific CD4^+^ T cells did not mediate these effects. We show these data in Figure 1 and describe them in the subsection “Stress conditions cause a severe phenotype of EAE”.

*2) More supporting evidence would be useful for implicating the HPA axis, which is currently done on the basis of higher blood aldosterone levels in stressed versus unstressed mice.*

We have added data of cortisol levels in serum as evidence for the involvement of the HPA axis. We show these data in Figure 1—figure supplement 1 and describe them in the subsection “Stress conditions cause a severe phenotype of EAE” and in the first paragraph of the subsection “The PVN-meditated sympathetic pathway is involved in the development of brain micro-inflammation at specific vessels under stress condition”.

*3) It is not clear that the combination of chronic stress and autoimmune CD4 T cells causes 'atypical EAE', since no data regarding the EAE itself are provided. In fact, the manuscript does not provide any information on whether the L5-specific point of initial autoimmune inflammation they have identified earlier is ALSO operative in chronically stressed recipients leading to an unaltered clinical course of neurological defects, or whether that entry point is no longer used, leading a far more radical restructuring of the clinical syndrome.*

Thank you for the comment. We have added more details to support our claim. Chronic stress caused pathogenic CD4^+^ T cells and MHC class II-positive cells to accumulate at specific vessels of the boundary region of the third ventricle region, thalamus, and dentate gyrus, but not at the L5 cord vessels (Figure 3). Furthermore, EAE mice under stress had no paralyzed tail, uneven gait, or paralyzed rear leg, which are all normally observed in EAE mice without stress. These data are consistent with the L5 cord vessels not acting as a gateway for immune cells under chronic stress conditions. We describe these observations in the first paragraph of the subsection “Brain micro-inflammation at specific vessels is developed in a manner dependent on chemokines and inflammatory cytokines after pathogenic T cell transfer under stress condition”.

*Further, the finding that lansoprazol treatment completely eliminates clinical signs may be an indication that EAE itself may not be prominent in this complex situation of stress and autoimmunity.*

Thank you for the suggestion. Based on our findings, we hypothesized that EAE induction itself is induced even after lansoprazol treatment under stress. We found that pathogenic CD4^+^ T cells and MHC class II-positive cells accumulated comparably at the brain specific vessels independent of lansoprazol treatment in mice with pathogenic T cell transfer under stress condition. These data suggested that chronic stress changes the position of the gateway for immune cells from L5 cord vessels to brain specific vessels, while pathogenic T cells are activated even after lansoprasol treatment. In addition, we showed that lansoprazol treatment did not affect the accumulation of immune cells in L5 and EAE symptoms in mice without stress. All these data suggested that lansoprazol treatment does not affect the EAE induction and that the stress-gateway reflex alters the gateway of pathogenic T cells. These data are shown in Figure 3—figure supplement 1 and described in the first paragraph of the subsection “Brain micro-inflammation at specific vessels is developed in a manner dependent on chemokines and inflammatory cytokines after pathogenic T cell transfer under stress condition”.

*4) While the CNS studies are very thorough, there is little information describing their functional linkage to the stomach epithelial destruction, and about how stress 'primes' the stomach for receiving the nerve signals that initiate the epithelial damage.*

It has been reported that under very strong stress, the activation of p38 and MAPKAPK2 contributes to stomach epithelial destruction under muscarinic M3 receptor signaling via acetylcholine derived from vagus nerve (Debas & Carvajal, 1994; Jia et al., 2007; Uwada et al., 2017). These reports further showed that p38 inhibitors could suppress the stomach epithelial destruction under very strong stress. We investigated the activation status of p38 and MAPKAPK2 in stomach isolated from mice with pathogenic T cell transfer under our stress condition and detected phosphorylated p38 and phosphorylated MAPKAPK2 in the gastric mucosa. These results suggested tissue damage in the stomach and the upper level of the intestine due to p38 and MAPKAPK2 activation in mucosal tissues. These data are shown in Figure 2—figure supplement 1 and described in the first paragraph of the subsection “Gastrointestinal failure is induced by pathogenic T cell transfer under stress”.

*5) The manuscript provides no explanation of the relevant molecular pathway distinctions between acute stress versus chronic stress that lead to the latter, but not the former, causing the disease phenotype addressed.*

We have shown that increased CCL5 expression at the specific vessels is induced by a neural pathway that involves the PVN, the stress nucleus. Therefore, we compared CCL5 expression at the specific vessels between mice with acute or chronic stress, finding much more CCL5 expression in the latter, which is consistent with another report (Ramot et al., 2017). Based on these data, we suggest that acute stress is insufficient to induce the nerve activation in PVN that elicits CCL5 expression at the specific vessels. We show these data in Figure 4—figure supplement 2 and describe them in the second paragraph of the subsection “Brain micro-inflammation at specific vessels is developed in a manner dependent on chemokines and inflammatory cytokines after pathogenic T cell transfer under stress condition” and in the last paragraph of the subsection “The PVN-meditated sympathetic pathway is involved in the development of brain micro-inflammation at specific vessels under stress condition”.

*6) The manuscript provides no insights into the relevant distinctions that make CX3CL1 the primary neuro-immune gateway chemokine at L5 in earlier work from the authors, and CCL5 the gateway chemokine in the present data.*

As the reviewer suggested, we previously identified CX3CL1 as a key chemokine for the accumulation of activated monocytes at L5 ventral vessels and EAE relapse after pain induction (Arima et al., 2015). In this context, CX3CL1 is produced from the monocytes themselves in response to noradrenaline such that they accumulate at the L5 ventral vessels. A major distinction between the previous and the current study is that in the previous study the activated monocytes were already presented in the CNS as CNS-resident activated monocytes that had high CX3CR1 expression and infiltrated the CNS during the initial phase of EAE. The pain-mediated neural pathway then induced the accumulation of these CNS-resident activated monocytes at the L5 ventral vessels, which is where norepinephrine was produced via the pain-specific sympathetic pathway, followed by EAE relapse and the accumulation of other immune cells including pathogenic T cells from the blood stream. However, in the current study there are no CNS-resident activated monocytes in mice under stress. Therefore, CX3CL1, which is a critical chemokine for CNS-resident monocytes, is not critical for the stress-gateway reflex in this study. On the other hand, during the stress-gateway reflex, we recognized CCL5 is required for the recruitment of pathogenic CD4^+^ T cells, including Th1 and Th17 cells, to CNS-specific vessels from the periphery to trigger micro-inflammation and stimulate the subsequent recruitment of MHC class IIhiCD11b+ cells. (These MHC class IIhiCD11b+ cells are possibly similar or the same cell types as the activated monocytes, although activated monocytes from the periphery initially express low CX3CR1 in the CNS). We assume that the recruitment of MHC class IIhiCD11b+ cells may be at least partially dependent on CX3CL1, which itself is induced by cytokines from pathogenic T cells that accumulated at the specific vessels in response to CCL5.

We describe these findings in the second paragraph of the Discussion.

*7) The manuscript does not address whether the IFNgamma and IL-17 cytokines, relevant for the pathway shown, are necessarily made by the pathogenic autoreactive CD4 T cells.*

According to the reviewer’s suggestion, we performed additional experiments using IL-17A- or IFN-γ -deficient mice. We prepared pathogenic CD4^+^ T cells derived from these mice and transferred them into mice with stress. We showed that the accumulation of IL-17A- or IFN-γ -deficient pathogenic CD4^+^ T cells was significantly reduced at the specific blood vessels under stress conditions and resulted in less mortality. These data suggested that both IFNγ and IL-17A are necessary for the accumulation at specific vessels and the severe phenotypes. We show these data in Figure 4—figure supplement 3 and describe them in the last paragraph of the subsection “Brain micro-inflammation at specific vessels is developed in a manner dependent on chemokines and inflammatory cytokines after pathogenic T cell transfer under stress condition”.

*8) The manuscript does not show whether the role of local CCL5 is merely as a gateway for inflammation, in which case the pathological consequences of local injection either T cells or cytokines would be expected to be CCL5-independent.*

According to the reviewer’s suggestion, we investigated the mortality of mice with microinjections of cytokines at the specific vessels under stress condition in the presence or absence of anti-CCL5 antibody treatment and found anti-CCL5 antibody treatment had no significant effect. This result suggested that CCL5 mainly contributes to the accumulation of immune cells at the specific vessels, while the effect of cytokine injection is CCL5-independent. We show these results in Figure 4—figure supplement 4 and describe them in the last paragraph of the subsection “Brain micro-inflammation at specific vessels is sufficient to induce fatal gastrointestinal failure under stress condition”.

*9) The manuscript does not address the cellular provenance of ATP sensing (is the ATP receptor needed on immune lineage cells and/or on neural lineage cells, for example?), and more broadly, it does not provide any positioning of ATP sensing in the pathogenic pathway it identifies (is it upstream or downstream of CCL5, IFNγ/IL17, neuronal activation…).*

In the previous draft, we showed that directly microinjected ATP at the specific vessels resulted in gastrointestinal disease and subsequent high mortality under stress condition (Figure 4) and the blockade of ATP signaling suppressed severe phenotypes in mice with cytokine microinjection under stress condition (Figure 6), suggesting that ATP is downstream of cytokine stimulation and acts as a neural stimulator. In the revised manuscript, we have added immunohistochemical staining results. Neurons (Neurofilament L-positive cells) around the specific vessels expressed an ATP receptor, P2RX7, and after cytokine or ATP stimulation neurons in the DMH region also showed NFATc1 activation, which is a downstream signaling component of P2RX7 (Grol, Pereverzev, Sims, & Dixon, 2013). Thus, we suggested that ATP can be sensed by neurons that connect specific vessels to the DMH region in a manner dependent on P2RX7 expression. In combination with our responses to Comments #7 and #8, we propose the following pathogenic pathway: CCL5, which is expressed from endothelial cells at the specific vessels, followed by the accumulation of pathogenic CD4^+^ T cells → IFNγ/IL17, which is expressed from pathogenic CD4^+^ T cells, followed by enhanced activation of endothelial cells at the specific vessels → ATP, which is expressed from the activated endothelial cells at the specific vessels → the activation of neurons in the DMH region, which are distributed around the specific vessels. We show these data in Figure 6—figure supplement 1 and describe them in the subsection “ATP induced at the sites of brain micro-inflammation activates the PVN/DMH and severe gastrointestinal failure”.

10) In fact, it would be very useful to have a cartoon-model of the pathway to provide an overview of the connections being made.

We added a cartoon-model of the pathway in Figure 7 and described them in the fourth paragraph of the Discussion.

*11) As a textual point, the manuscript perhaps makes too many MS-specific claims; EAE may not be a sufficiently robust MS model for that. The data are remarkable enough independent of what they may say about MS.*

According to the reviewer’s comment, we added the possible involvement of this pathway in other diseases including neurodegenerative diseases, dementia and psychological disorders in the sixth paragraph of the Discussion.

*12) It would be very useful to have the 'data not shown' to be shown at least in supplementary figures.*

According to the reviewer’s comment, data not shown are presented in Figure 2—figure supplement 2, Figure 3—figure supplement 2, Figure 4—figure supplement 1, Figure 4—figure supplement 5, and Figure 7—figure supplement 1.

[Editors' note: further revisions were requested prior to acceptance, as described below.]

*General comments:*

*The authors speak about pathogenic T cells throughout the manuscript, without even relating to CD8 cells that are very much associated with EAE.*

We sorted CD4^+^ T cells from mice EAE induced with a MOG peptide plus CFA (active EAE model) and stimulated the cells with a MOG peptide in the presence of MHC class II+ antigen presenting cells and cytokines in vitro followed by intravenously injecting them into wild type C57BL/6 mice (the transfer EAE model). Therefore, there are no CD8^+^ T cells in the pathogenic T cell population. Accordingly, we have changed the term “pathogenic T cells” to “pathogenic CD4^+^ T cells” for clarity (Introduction, first paragraph).

*Moreover, based on current models, IFN-γ does not have a major role in the pathology.*

There are many papers that describe the involvement of IFNγ particularly from pathogenic CD4^+^ T cells including Th1 and activated Th17 cells for the development of EAE ^1-5^. These papers are cited in the text (Introduction, first paragraph).

*The authors also describe resident monocytes in the brain. There are no resident monocytes in the brain, under any circumstances. The only resident myeloid cells in the brain are microglia, even under pathology; in EAE, the infiltrating monocytes should be termed monocyte-derived macrophages, and unlike the resident microglia, they do not express CX3CR1+.*

As explained in the answer to General comment #1, we have been using transfer EAE, which shows transient clinical symptoms, as our model. We called EAE mice that stopped showing clinical symptoms EAE-recovered mice. There is no relapse in EAE-recovered mice under normal condition. On the other hand, over 90% of MS patients experience relapse, and there are reports showing that the occurrence of pain is associated with neurologic symptoms and disease severity in patients with MS ^6-8^. We therefore hypothesized that pain sensation might induce MS relapse via a reflex pathway. Indeed, pain sensation induced EAE relapse in EAE-recovered mice. Mechanistic analysis showed that the CNS conditions between wild type and EAE-recovered mice are completely different even though both groups show no clinical symptoms. In the EAE-recovered mice, many MHC class IIhiCX3CR1+CD11b+ cells are found in the CNS. These cells are not activated microglial cells but activated monocytes coming from the periphery. The identification of the periphery as the source of MHC class IIhiCX3CR1+CD11b+ cells was done by two methods: parabiosis and labeling microglial cells with CX3CR1-Tdtomato. Importantly, these activated monocytes in the CNS of EAE-recovered mice stayed for a long time (over 150 days) after pathogenic CD4^+^ T cell transfer and are critical for the development of pain-mediated EAE relapse. We published these data in *eLife* in 2015. On the other hand, many papers have reported resident T cells stay in non-immune tissues for a very long time (over 150 days in some cases) after the initial immune response in the tissues and affect subsequent immune responses ^9,10^. The characteristics of these resident T cells and our activated monocytes in the CNS of the EAE-recovered mice are quite similar: (1) infiltrate the initially affected non-immune tissues, (2) infiltrate a long time, and (3) influence subsequent immune responses. Therefore, we describe the activated monocytes in the CNS of EAE-recovered mice as resident monocytes (Discussion, third paragraph).

*Specific comments:*

*1) The authors were requested to add a control consisting of autoimmune cells with non-CNS specificity – Ova specific T cells are not an appropriate control; the question is whether autoimmune T cells directed against self-antigens in areas outside the brain will affect the gut.*

We employed another transfer method, experimental autoimmune uveoretinitis (EAU), which is a model of uveitis. Wild type C57BL/6 mice were immunized with a retina antigen peptide, IRBP (1-20), with CFA followed by the induction of active type of EAU. We purified CD4^+^ T cells from mice with EAU symptoms followed by culturing with IRBP (1-20) in the presence of MHC class II+ antigen-presenting cells plus cytokines. The resulting retina-specific activated CD4^+^ T cells were intravenously injected into wild type C57BL/6 mice with stress to investigate whether gastrointestinal diseases developed. Although retina-specific activating CD4^+^ T cells accumulated in the eyes, no gastrointestinal diseases occurred. We show these data in Figure 1—figure supplement 1 and describe them in the subsection “Stress conditions cause a severe phenotype of EAE”.

*2) The authors were asked to add results relating to the HAPaxis; the additional measurements of cortisol do not help. Such experiments provide a correlation but not causal results. Blocking of the corticosteroid receptor or its function could add to our understanding of the mechanism.*

We performed the experiments the reviewer suggested. We employed two antagonists of the corticosteroid receptor such as mifepristone and guggulsterone and treated them in mice after microinjection of cytokines at the specific vessels under stress. We found that cytokine-mediated gastrointestinal diseases were not suppressed by the antagonist treatments. These results suggested that the HPA axis has a minimum role in the development of micro inflammation-mediated gastrointestinal diseases under stress. We show these data in Figure 1—figure supplement 3 and describe them in the sixth paragraph of the Discussion.

*3) No mechanistic explanation is provided for the changes in the autoimmune T cell gateway from L5 to the brain.*

We previously showed that the L5-gateway is established by specific neural activation via gravity-mediated soleus stimulation ^11^. Because we found that mice with stress show reduced movement and often lie down, we hypothesize they have reduced stimulation of the soleus muscles, which are the main anti-gravity muscles. The sensory neurons from soleus muscles connect to L5 dorsal root ganglions (DRG), which are activated by anti-gravity responses of the soleus muscles in wild type mice. Consistent with our hypothesis, we found that mice with stress reduce L5 DRG activation and chemokine expression from L5 dorsal vessels (L5 gateway). We show these data in Figure 3—figure supplement 3 and describe them in the second paragraph of the Discussion.

*4) No explanation is given for the linkage between the T cells and activation of p38.*

It is known that pathogenic CD4^+^ T cells for the transfer EAE express cytokines including IFN-γ, IL-17A, and IL-6 ^12^, and very strong stress alone induces gastric disease via p38 activation at the affected tissues by the activation of the vagal pathway ^13-15^. We showed that the injection of IFN-γ, IL-17A, and IL-6 or ATP at specific vessels in the brain of mice with relatively low stress establishes a neural pathway via DMH and DMX (nucleus of vagal nerves) followed by the development of fatal gastrointestinal diseases (Figure 4 and Figure 5) and the activation of p38 in stomach (Figure 2—figure supplement 1). Moreover, cytokine stimulation in endothelial cells causes the expression of ATP (Figure 6). These results strongly suggested that cytokines from pathogenic CD4^+^ T cells act on endothelial cells at the specific vessels and induce ATP around the specific vessels. The resulting ATP activates the DMH-vagal axis, which is important for the development of gastrointestinal diseases as well as the activation of p38. Therefore, cytokines from the pathogenic CD4^+^ T cells are a triggering factor to activate p38 in the affected tissue via DMH-vagal axis activation in mice with relatively low stress. We describe these findings in the seventh paragraph of the Discussion.

*5) The role of Cx3CL1 expression by monocytes is puzzling; the receptor is normally expressed by patrolling monocytes or within the CNS by microglia.*

As described in our answer to General comment #3, resident monocytes with CX3CR1 in EAE-recovered mice accumulate from the periphery to stay in the CNS for a long time and contribute to the development of pain-mediated EAE-relapse. Therefore, it is possible that the origin of the resident monocytes could be patrolling monocytes, as suggested by the reviewer. We previously showed that resident monocytes with CX3CR1 express CX3CL1 after stimulation with norepinephrine, which is produced around the ventral vessels of the spinal cord following activation of the pain-specific neural pathway ^16^. These results suggested a CX3CR1-CX3CL1 autocrine loop in the resident monocytes that is regulated by norepinephrine. We add these explanations in the fourth paragraph of the Discussion.

References:

1) Goverman, J. Autoimmune T cell responses in the central nervous system.

Nature reviews. Immunology 9, 393-407, doi:10.1038/nri2550 (2009).

2) Jager, A., Dardalhon, V., Sobel, R. A., Bettelli, E. & Kuchroo, V. K. Th1, Th17, and Th9 effector cells induce experimental autoimmune encephalomyelitis with different pathological phenotypes. Journal of immunology 183, 7169-7177, doi:10.4049/jimmunol.0901906 (2009).

3) El-behi, M., Rostami, A. & Ciric, B. Current views on the roles of Th1 and Th17 cells in experimental autoimmune encephalomyelitis. Journal of neuroimmune pharmacology: the official journal of the Society on NeuroImmune Pharmacology 5, 189-197, doi:10.1007/s11481-009-9188-9 (2010).

4) Duhen, R. et al. Cutting edge: the pathogenicity of IFN-gamma-producing Th17 cells is independent of T-bet. Journal of immunology 190, 4478-4482, doi:10.4049/jimmunol.1203172 (2013).

5) Ottum, P. A., Arellano, G., Reyes, L. I., Iruretagoyena, M. & Naves, R. Opposing Roles of Interferon-Gamma on Cells of the Central Nervous System in Autoimmune Neuroinflammation. Front Immunol 6, 539, doi:10.3389/fimmu.2015.00539 (2015).

6) Ehde, D. M. et al. Chronic pain in a large community sample of persons with multiple sclerosis. Mult Scler. 9, 605-611 (2003).

7) Ehde, D. M., Osborne, T. L., Hanley, M. A., Jensen, M. P. & Kraft, G., H. The scope and nature of pain in persons with multiple sclerosis. Mult Scler. 12, 629-638 (2006).

8) O'Connor, A. B., Schwid, S. R., Herrmann, D. N., Markman, J. D. & Dworkin, R. H. Pain associated with multiple sclerosis: systematic review and proposed classification. Pain 137, 96-111, doi:10.1016/j.pain.2007.08.024 (2008).

9) Iijima, N. & Iwasaki, A. T cell memory. A local macrophage chemokine network sustains protective tissue-resident memory CD4 T cells. Science 346, 93-98, doi:10.1126/science.1257530 (2014).

10) Mueller, S. N. & Mackay, L. K. Tissue-resident memory T cells: local specialists in immune defence. Nature reviews. Immunology 16, 79-89, doi:10.1038/nri.2015.3 (2016).

11) Arima, Y. et al. Regional neural activation defines a gateway for autoreactive T cells to cross the blood-brain barrier. Cell 148, 447-457, doi:S0092-8674(11)00088-8 [pii]

10.1016/j.cell.2012.01.022 (2012).

12) Langrish, C. L. et al. IL-23 drives a pathogenic T cell population that induces autoimmune inflammation. J Exp Med. 201, 233-240 (2005).

13) Debas, H. T. & Carvajal, S. H. Vagal regulation of acid secretion and gastrin release. Yale J Biol Med 67, 145-151 (1994).

14) Jia, Y. T. et al. Activation of p38 MAPK by reactive oxygen species is essential in a rat model of stress-induced gastric mucosal injury. Journal of immunology 179, 7808-7819 (2007).

15) Uwada, J. et al. Activation of muscarinic receptors prevents TNF-alpha-mediated intestinal epithelial barrier disruption through p38 MAPK.

Cell Signal 35, 188-196, doi:10.1016/j.cellsig.2017.04.007 (2017).

16) Arima, Y. et al. A pain-mediated neural signal induces relapse in murine autoimmune encephalomyelitis, a multiple sclerosis model. eLife 4, e08733, doi:10.7554/eLife.08733 (2015).